# READ-SQL: Reasoning Path Decomposer for Text-to-SQL

## Abstract

Text-to-SQL is a longstanding task aimed at automatically converting natural language questions into SQL queries for database retrieval. Despite impressive advancements, particularly with Large Language Models (LLMs), existing methods still struggle with issues such as misinterpreted, omitted, or unwanted constraints. To address these challenges, we propose **READ-SQL**, a novel framework employing a reasoning path dcomposer, **READ**ER, for text-to-**SQL** tasks. READER decomposes SQLs into clauses, sub-SQLs, and reasoning paths, supporting data preparation and confidence level determination in post-processing. READ-SQL comprises two main models: a Generator and a Corrector, both trained via LoRA for parameter efficiency. Based on READER's decomposition, READ-SQL generates two types of augmented data using an LLM: question/SQL pairs and question/reason pairs. The Generator is trained on both original and augmented data to identify constraint changes and enhance reasoning. The Corrector is trained on data from READER's post-processing, improving self-correction by refining high-confidence SQLs and addressing low-confidence elements. Extensive experiments show that READ-SQL significantly outperforms leading baselines, with READ-SQL-3B achieving 57.37% execution accuracy on BIRD's Dev set, surpassing several 7B-parameter models and setting a new state-of-the-art with fewer parameters. Additionally, READER and the Corrector show broad applicability when integrated with LLMs or other base models.

## 1 Introduction

Text-to-SQL, a longstanding and pivotal task in natural language processing, focuses on transforming natural language questions into executable SQLs [1], streamlining database interactions for non-experts and significantly enhancing information retrieval efficiency (Deng et al., 2022; Katsogiannis-Meimarakis & Koutrika, 2023; Liu et al., 2024a). Recent advances in task decomposition, intermediate representations, and post-processing strategies have significantly pushed the field forward (Wang et al., 2023; Guo et al., 2019; Pourreza & Rafiei, 2023). Additionally, the integration of Large Language Models (LLMs) has greatly enhanced natural language understanding and SQL generation through broader context and carefully crafted prompts (Wang et al., 2023; Li et al., 2024a; Talaei et al., 2024b). However, challenges remain in applying these advancements to real-world scenarios, particularly when handling vague questions in large, complex database schemas (Zhang et al., 2024; Liu et al., 2024b; Li et al., 2024a; Liu et al., 2024a).

Figure 1 illustrates three typical categories of errors in text-to-SQL, with JOIN-related errors classified accordingly. We attribute these errors to the inherent disparity between natural language and the (semi-)structured syntax of SQL (Liu et al., 2024a):

- **Misinterpreted constraints:** These occur when the model incorrectly parses the natural language query, resulting in erroneous SQL clause (Li et al., 2023c; Talaei et al., 2024b).
- **Omitted constraints:** These arise when the model overlooks essential elements, leading to incomplete SQLs (Pourreza & Rafiei, 2023; Talaei et al., 2024b; Wang et al., 2023).
- **Unwanted constraints:** These involve superfluous clauses that exceed the requirements of the natural language input (Talaei et al., 2024b; Wang et al., 2023).

---

[1] For brevity, we use "SQLs" to refer to SQL queries and "sub-SQLs" to refer to sub-SQL queries.

Figure 1: Typical errors in text-to-SQL: Texts with yellow shading highlight constraints in the questions, while red shading marks incorrect SQL clauses, and blue shading suggests constraint modifications. The error proportions in CodeS (Li et al., 2024c) are detailed in Appendix A.5.2.

Existing methods address these challenges from two main perspectives: (1) **SQL-like grammar languages**: These approaches reduce the complexity of SQL generation by employing intermediate SQL-like grammar representations (Gan et al., 2021c; Yu et al., 2018a; Eyal et al., 2023). They can be easily integrated with pre-trained models and large language models (LLMs) to produce effective results (Gan et al., 2021c; Pourreza & Rafiei, 2023; Li et al., 2023a; Rai et al., 2023). (2) **Direct output of SQL structure**: These methods generate the structure of SQL directly, leveraging grammar information as an intermediate representation and focusing on the syntactic structure of SQL (Gu et al., 2023b;a; Yu et al., 2018a). However, existing methods primarily focus on optimizing SQL generation without improving the model's understanding of the question or establishing strong relationships between questions and SQL clauses. Additionally, since these methods do not employ an end-to-end architecture, there is potential for information loss during the process (Liu et al., 2024a).

To address these shortcomings, we propose **READ-SQL**, a novel framework comprising two key models, a Generator and a Corrector, both supported by a reasoning path decomposer, **READ**ER, specifically designed for text-to-**SQL** tasks. READER can parse an executable SQL into an Abstract Syntax Tree (AST) (Wang et al., 1997) and decompose it into clauses, forming sub-SQLs and reasoning paths. READ-SQL generates two types of augmented data via an LLM: question/SQL pairs and question/reason pairs, embedding information about subtle constraint changes and the connection between questions and reasoning paths. The Generator is trained on a multi-task fine-tuning framework via LoRA (Hu et al., 2022), leveraging both the original and augmented data, and outputs either SQLs or reasons (describing sub-SQLs generation) based on prefix tokens. We aim to enhance the model's understanding of questions by incorporating additional question/SQL pairs and bridging the gap between questions and SQL through question/reason pairs. For post-processing, READ-SQL employs a Corrector, also trained via LoRA, on the Generator's processed output SQLs, which are categorized by READER into two types: high-confidence clauses for basic SQLs and low-confidence clauses for retrieving similar items from table schema. The Corrector is trained on these two types of data to produce the final SQLs, enabling precise self-correction.

We summarize our main contributions as follows:

- We propose READ-SQL, featuring two key models: a Generator and a Corrector. The Generator captures subtle differences between questions and SQLs and enhances the connection between questions and reasoning paths, improving constraint recognition and reasoning abilities. The Corrector refines the Generator's processed SQL outputs for precise self-correction.
- Both the Generator and Corrector rely on READER, which decomposes SQLs into clauses to form sub-SQLs and reasoning paths. This aids in data generation for the Generator and refines its outputs for the Corrector.
- Extensive experiments show that READ-SQL outperforms leading baselines, with READ-SQL-3B achieving 57.37% execution accuracy on the BIRD Dev, surpassing several 7B-parameter models. Additionally, READER and the Corrector are highly versatile, making them suitable for integration with LLMs or other base models. Furthermore, READ-SQL significantly reduces three common types of text-to-SQL errors.

## 2 RELATED WORK

Generating accurate SQLs from natural language questions, commonly known as text-to-SQL, is an active area of research within both the natural language processing and database communities (Liu et al., 2024a; Zhang et al., 2024). In the following, we review two mainstreams of related work.

### 2.1 TEXT-TO-SQL WITH PRE-TRAINED LANGUAGE MODELS

The development of text-to-SQL has progressed from early neural network-based methods, such as IRNet (Guo et al., 2019) and Bridge (Lin et al., 2020), to pre-trained model approaches like RESD-SQL (Li et al., 2023a), and now to the current era of large models, including DIN-SQL (Pourreza & Rafiei, 2023) and CHESS (Talaei et al., 2024b). Throughout this evolution, task decomposition has remained a fundamental process in text-to-SQL, typically involving schema linking, SQL generation, and post-processing. Various methods have been proposed to handle these sub-tasks:

- **Schema linking** is a critical task that involves identifying the relevant database table columns and values referenced in natural language questions. Pre-trained models (Li et al., 2023a; 2024c) and LLMs have significantly improved schema linking performance (Talaei et al., 2024b; Pourreza & Rafiei, 2023). Even when irrelevant table schema is present, LLMs can accurately generate SQLs (Maamari et al., 2024). Schema linking can help models alleviate the gap between question and SQL, but even state-of-the-art methods (Li et al., 2023a; 2024c) suffer from type three errors in text-to-SQL. In this paper, we adopt the schema linking solution from (Li et al., 2024c) and focus on the remaining two steps, exploring potential ways to better understand the intent behind questions during SQL generation and post-correction.
- **SQL generation** involves the challenge of bridging the gap between the flexibility of natural language and the rigid structure of SQLs. This discrepancy introduces two key issues: the model may misinterpret the query's intent or generate incorrectly formatted SQL. Current approaches often rely on LLMs or fine-tuned pre-trained models for end-to-end solutions (Talaei et al., 2024b; Li et al., 2024c). Some researchers explore intermediate representations, like Natural SQL, to simplify generation (Pourreza & Rafiei, 2023), but these methods have limited practical application and do not significantly improve query comprehension. Other approaches, such as (Ye et al., 2023), decompose the task into sub-problems, generating sub-SQLs before forming the final SQLs. While they offer a multi-step reasoning solution, they carry the risk of error propagation. The limitations of existing methods motivate us to design an end-to-end framework that eliminates error propagation, enhancing multi-step reasoning and question comprehension.
- **Post-processing** aims to refine generated SQLs to improve both user satisfaction and query accuracy. DIN-SQL (Pourreza & Rafiei, 2023) introduces a self-correction module to detect potential syntax errors, while MAC-SQL (Wang et al., 2023) employs a multi-agent approach for error identification and correction. In contrast, C3 (Dong et al., 2023) and DAIL-SQL (Gao et al., 2024) apply self-consistency by sampling multiple results and selecting the most consistent one. CodeS (Li et al., 2024c) selects the first executable SQL as the result. Bertrand-DR (Kelkar et al., 2020) reorders sampled results to align with user preferences. However, these methods often struggle to accurately identify where corrections are needed or focus solely on selecting outputs without modifying the SQL itself. These limitations motivate us to develop a more effective approach for identifying potential error clauses and systematically generating the final SQL.

### 2.2 TEXT-TO-SQL WITH ABSTRACT SYNTAX TREES

The Abstract Syntax Tree (AST) (Wang et al., 1997) is a tree-like data structure that represents the syntactic structure of source code, enabling compilers and other tools to efficiently parse and analyze code. Each node in an AST corresponds to a structural element, such as an operator or function call. Decoders like RAT-SQL (Wang et al., 2020) and IRNet (Guo et al., 2019) utilize a tree-structured approach to generate the AST for a SQL query and then convert it back into SQL, ensuring grammatical accuracy. ASTormer (Cao et al., 2023) builds on this by utilzing an AST-aware transformer decoder that incorporates grammatical structure with both absolute and relative position encoding. These methods fall under the category of grammar-based decoders (Zhang et al., 2024). However, with the rise of LLMs (Wei et al., 2022a), the focus has shifted away from this approach. We revisit the use of ASTs to ensure their relevance in the LLM era. By decomposing ASTs, we can derive sub-SQLs and trace their evolution, facilitating the model's step-by-step reasoning.

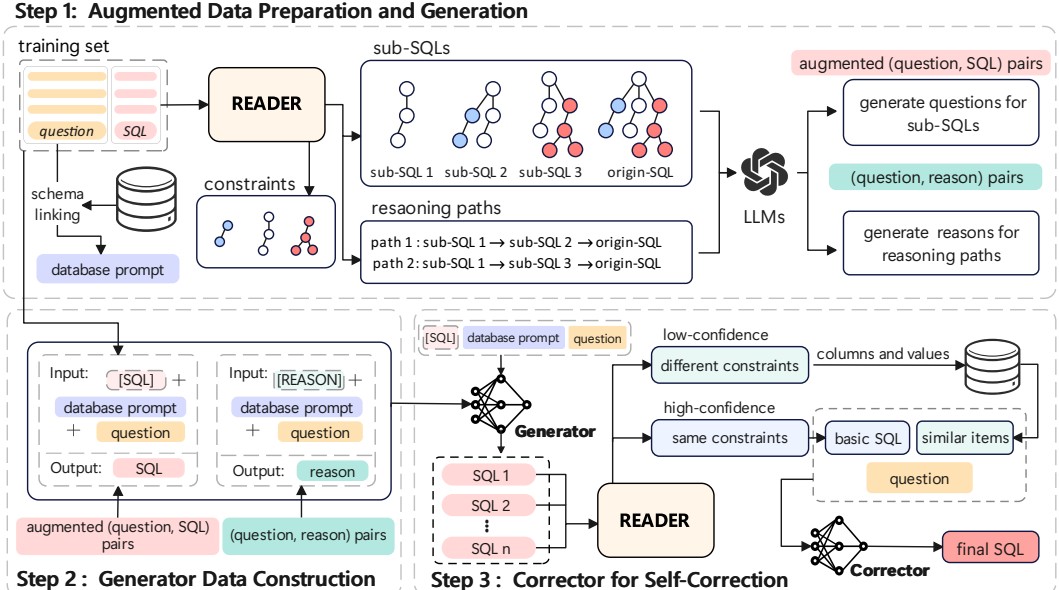

Figure 2: The three main steps in READ-SQL: **READER** serves as the core module, with **Generator** and **Corrector** as two models to generate SQLs. Further details are provided in the main text.

## 3 METHODOLOGY

**Problem Definition**   Formally, given a natural language question $Q$ and a database $\mathcal{D}$ with schema $\mathcal{S}$, the text-to-SQL task aims to translate $Q$ into a SQL query $y$ that can be executed on $\mathcal{D}$ to answer the question $Q$. The database $\mathcal{D}$ contains the schema $\mathcal{S} = (\mathcal{T}, \mathcal{C}, \mathcal{R})$ of three components: a set of $N$ tables $\mathcal{T} = \{t_1, t_2, ..., t_N\}$; a set of columns $\mathcal{C} = \{c_1^1, ..., c_{n_1}^1, ..., c_1^N, ..., c_{n_N}^N\}$ associated with the tables, where $n_i$ is the number of columns in the $i$-th table; and a set of foreign key relations $\mathcal{R} = \{(c_k^i, c_h^j) \mid c_k^i, c_h^j \in \mathcal{C}\}$, where $(c_k^i, c_h^j)$ indicates a foreign key relationship between two columns. We use $M = \sum_{i=1}^{N} n_i$ to denote the total number of columns in $\mathcal{D}$.

**Architecture.**   Figure 2 illustrates the three main steps in READ-SQL: (1) Preparing and generating the augmented data, (2) constructing the training data for the Generator, and (3) constructing the training data for the Corrector. At the core of READ-SQL is the READER module. We will elaborate on them one-by-one in the following.

### 3.1   READER: A REASONING PATH DECOMPOSER

READER parses executable SQL into constraints, generating sub-SQLs and reasoning paths. This forms the foundation for enhanced data generation and refined self-correction. Figure 3 illustrates the four main steps of READER: (1) parsing an SQL into AST; (2) identifying all constraints from the AST; (3) get sub-SQLs by removing constraints on AST and (4) construct reasoning paths for the sub-SQLs. Further details can be found in Appendix A.1. It is noted that

- **Step 2: Identify constraints.** A constraint is defined as a sub-tree in the AST where the root node is an operation type and all its child nodes are non-operation types (see Appendix A.1.2 for a detailed explanation). For instance, the "SELECT" node represents an operation, while its child, "name", is a non-operation node. Similarly, the "WHERE" node and its corresponding child are recognized by READER as a constraint, indicated by their background color in Figure 3.
- **Step 3 : Sequentially delete constraints.** READER gets sub-SQLs by removing constraints. READER initializes a binary tree to store results, with the root node storing the AST parsed from the original SQL. At each level, READER removes one constraint from the root, storing the remaining sub-tree as the right leaf node, while replicating the root in the left leaf node. For example, the circle labeled 3 in Figure 3 is obtained by removing the "WHERE age=18" constraint

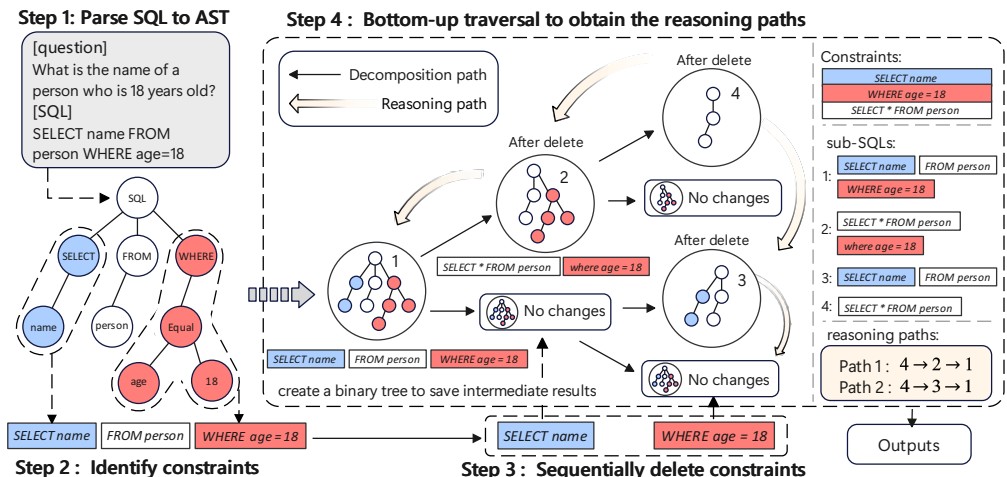

Figure 3: Illustration of READER for parsing an SQL into AST, forming sub-SQLs and reasoning paths; see details in the main text.

from the original AST and is stored in the right leaf of the root node. This process continues until all constraints are enumerated, and the sub-SQLs are stored in the leaves of the binary tree.

- **Step 4 : Obtain the reasoning paths.** READER starts from the rightmost leaf node, which represents the minimal sub-SQL with all constraints removed (e.g., "SELECT * FROM person" in Figure 3). READER then conducts a breadth-first search upward, identifying nodes that add one additional constraint compared to the current node, and incorporating them as the next points in the reasoning path. This process continues until reaching the root node. For example, Figure 3 outputs two reasoning paths, $4 \rightarrow 2 \rightarrow 1$ and $4 \rightarrow 3 \rightarrow 1$, where each node introduces one additional constraint compared to the previous node in the path.
- After processing the SQL, READER produces three outputs: a set of SQL clauses indicating the constraints from the SQL, the corresponding sub-SQLs, and the reasoning paths.

It is noted that READER shares similarities with DeSQL (Haroon et al., 2024) in its approach. However, READER extends beyond DeSQL by accommodating a wider range of SQL structures, deriving reasoning paths, and applying these processes to text-to-SQL tasks. We demonstrate READER's superiority over similar tools in the Appendix A.1.4.

## 3.2 AUGMENTED DATA GENERATION

After obtaining results from READER, we utilize an LLM to generate two types of augmented data:

- **Augmented (question, SQL) pairs**: A question is generated from a sub-SQL using a prompt template, as shown in Figure 9. These augmented data enrich the model's sensitivity to constraints, helping it better detect implicit constraints in natural language.
- **(question, reason) pairs**: A reason is generated from a reasoning path using a prompt template, as shown in Figure 11, to describe the chain-of-thought (CoT) (Wei et al., 2022b) process involved in constructing sub-SQLs step-by-step. These (question, reason) pairs help the model understand the SQL construction process to enhance reasoning abilities.

## 3.3 GENERATOR FOR INITIAL SQLS GENERATION

The Generator is the first model to generate SQLs in READ-SQL. Given a table, we first construct the database prompt following Li et al. (2024c). Next, we format the input as:

- **Original and augmented (question, SQL) pairs**: $x_{\text{SQL}}$ is to concatenate the following tokens:

$$x_{\text{SQL}} = [\text{SQL}] + \text{database prompt} + \text{question} \tag{1}$$

- **(question, reason) pairs**: $x_{\text{R}}$ is to concatenate the following tokens:

$$x_{\text{R}} = [\text{REASON}] + \text{database prompt} + \text{question} \tag{2}$$

Figure 4: An example of self-correction inference by the Corrector. In the final input, the yellow background color indicates the data organization format for better visualization and understanding.

It is noted that the input of $x_{\text{SQL}}$ and $x_{\text{R}}$ differs only on the prefix token.

READ-SQL then employs a multi-task framework for supervised fine-tuning (Hsieh et al., 2023) on top of LoRA (Hu et al., 2022) to train the Generator using the above two kinds of data, by minimizing the following loss (representing the loss on a single instance):

$$L = - \sum_{i=1}^{|y^{\text{SQL}}|} \log(P_G(y_i^{\text{SQL}}|y_{<i}^{\text{SQL}}, x_{\text{SQL}})) - \lambda \sum_{i=1}^{|y^{\text{R}}|} \log(P_G(y_i^{\text{R}}|y_{<i}^{\text{R}}, x_{\text{R}}), \tag{3}$$

where $P_G$ represents the conditional probability of the Generator, $\lambda$ is a hyperparameter balanceing the loss on the (question, SQL) pairs and (question, reason) pairs.

It is worth emphasizing that the purpose of adding (question, reason) pairs during training is to help the Generator understand the reasoning process behind SQL and assist in its generation. In the inference stage, we only need the Generator to produce SQLs. We compare other fine-tuning methods in the appendix A.5.6, which are not as effective as the multi-tasking framework.

### 3.4 CORRECTOR FOR SELF-CORRECTION

Corrector uses READER to re-evaluate the low-confidence constraints in the results generated by the Generator, producing the final SQL, as illustrated in the Figure 2. To begin, we will explain the reasoning process of Corrector. This process primarily consists of two parts: constructing the input for Corrector, and generating the final SQL. The entire process is illustrated in Figure 4. Corrector's input relies on READER, which parses the SQL to extract two types of constraints:

- **High-confidence constraints** appear consistently in all generated SQLs. For example, READER identifies four identical constraints and combines them to form a basic AST, which is parsed to a basic SQL, as shown in Figure 4. Basic SQL retains the consistent part of the Generator's result, allowing Corrector to generate SQL based on it, which helps reduce errors.
- **Low-confidence constraints** are those that differ in other SQLs or appear in only some SQLs. Corrector requires additional information to generate accurate SQL from low-confidence clauses. First, it extracts the table names, column names, and values from the low-confidence clauses. Then, READ-SQL uses search engines, such as Elasticsearch, to retrieve related information, including table schema, foreign keys, and value matches.

Finally, READ-SQL organize basic SQL and the retrieved table schema, foreign keys and value matches into the final input, as shown in Figure 4, allowing Corrector to output the final SQL.

Similar to the Generator, the Corrector model is also fine-tuned using LoRA. However, since the Corrector requires the Generator's output, we perform cross-validation on the text-to-SQL training set to obtain the Generator's outputs for each fold. Following the previously described process, we construct the input data and combine the data from all folds to create the final training dataset for the Corrector. Detailed steps are provided in the appendix A.4.

## 4 EXPERIMENTS

### 4.1 EXPERIMENTAL SETTINGS

**Baselines**  For supervised fine-tuning, nearly all baselines are derived from the state-of-the-art (SOTA) text-to-SQL approaches listed on the official leaderboards of the BIRD and Spider benchmarks (Li et al., 2023a; 2024c; Yang et al., 2024; Li et al., 2023b; Scholak et al., 2021). We select SFT CodeS as our primary competitive baseline (Li et al., 2024c). Additionally, we compare READ-SQL with LLM-based methods (Pourreza & Rafiei, 2023; Gao et al., 2024).

**Datasets**  We conduct experiments on two English text-to-SQL benchmarks: BIRD (Li et al., 2023c) and Spider (Yu et al., 2018b). To evaluate the model's robustness, we also evaluate READ-SQL on three Spider variants: Spider-DK (Gan et al., 2021b), Spider-Syn (Gan et al., 2021a), and Spider-Realistic (Deng et al., 2021). BIRD includes 9,428 training samples and 1,534 development samples, while Spider has 8,659 training samples and 1,034 development samples, both featuring hidden test sets. Details of the datasets are provided in Appendix A.5.1.

**Metrics**  To evaluate the performance of the Text-to-SQL parser, following Li et al. (2024c); Talaei et al. (2024b); Pourreza & Rafiei (2023); Li et al. (2024a), we apply the following metrics: (1) For the BIRD benchmark, we employ execution accuracy (EX), which measures whether the generated SQL retrieves the correct results from the database, and the Valid Efficiency Score (VES), which is determined by dividing the execution time of the ground truth SQL query by the execution time of the predicted SQL query, to quantify the accuracy and performance efficiency of the generated SQLs. Since VES relies on the environment, we rerun the results of CodeS for a fair comparison. (2) For Spider and its variants, we adopt the test-suite accuracy (TS) score (Zhong et al., 2020), which measures whether generated SQLs consistently pass the EX evaluation across multiple database instances, thereby reducing false positives—instances where a prediction that is semantically different from the correct answer coincidentally matches the same denotation in a specific database.

**Implementation Details**  We provide a detailed explanation of READ-SQL's Generator and Corrector separately.

– **Generator**: We use GLM-4-0520 (Zeng et al., 2024) as the base LLM to generate questions and reasoning paths. For schema linking, we adopt the same method as CodeS (Li et al., 2024c) to obtain the database prompt. During training, we fine-tune the model using the LoRA (Hu et al., 2022) technique, with CodeS-1B and CodeS-3B as base models. The learning rate is set to 1e-4, the training runs for 6 epochs, and we use a batch size of 8 with a $\lambda$ value of 8. For inference, the beam size is set to 4, and greedy decoding is applied.
– **Corrector**: We utilize Elasticsearch as a retrieval tool to select the top 4 columns and the top 2 cell values based on approximate matching. When constructing the dataset, we employ four-fold cross-validation, training the Generator on three folds with the above parameters while evaluating on the remaining fold. This process is repeated across all four folds. The Corrector is also fine-tuned using LoRA (Hu et al., 2022) in CodeS-1B and CodeS-3B, with a learning rate of 1e-4, 6 epochs, and a batch size of 8. During inference, we set the beam size to 4 and select the first executable SQL as the final output.

**Environments**  All the experiments are run on a server with 8 NVIDIA RTX 3090 GPUs of 24 GB memory for the models and an AMD EPYC 7742 CPU of 128 GB memory for testing VSE. More information is detailed in Appendix A.5.3.

### 4.2 MAIN RESULTS

Table 1 reports the results of compared methods on the benchmarks, highlighting the following: (1) For the BIRD dataset (Dev set), READ-SQL achieves the best performance across all methods in both EX and VES. Notably, READ-SQL-3B improves upon SFT CodeS-3B in EX by 2.35% and even outperforms SFT CodeS-7B by 0.37%. It also significantly outperforms strong baselines like DIN-SQL and DAIL-SQL, which utilize GPT-4. Regarding VES, we observe that it is influenced by the computation environment, so we rerun SFT CodeS for a fair comparison. This results in higher VES values than those reported in (Li et al., 2024c). Nonetheless, READ-SQL consistently

Table 1: Performance comparison on BIRD and Spider benchmarks: "-/-" in the VES column indicates that the results are copied from the original paper and reproduced by us. Values in parentheses record READ-SQL improvement over SFT CodeS with the same model size.

| Methods | BIRD Dev | | Spider Dev | | Spider Test |
|---|---|---|---|---|---|
| | EX (%) | VES (%) | EX (%) | TS (%) | EX (%) |
| Prompting Methods w/ Closed-Source LLMs | | | | | |
| CHESS + GPT-4 (Talaei et al., 2024a) | **65.00** | - | - | - | **87.2** |
| PURPLE + GPT-4 (Ren et al., 2024) | - | - | **87.8** | **83.3** | - |
| PTD-SQL + GPT-4 (Luo et al., 2024) | 57.0 | 57.7/- | 85.7 | - | - |
| SuperSQL + GPT4 (Li et al., 2024b) | 58.5 | 61.99/- | 87.0 | - | - |
| DIN-SQL+GPT-4 (Pourreza & Rafiei, 2023) | 50.72 | 58.79 / - | 82.8 | 74.2 | 85.3 |
| DAIL-SQL + GPT-4 (Gao et al., 2024) | 54.76 | 56.08 / - | 83.1 | 76.6 | 86.6 |
| Fine-tuning Models w/ Open-Source LLMs | | | | | |
| RESDSQL-3B + NatSQL (Li et al., 2023a) | 43.9 | 45.64 / - | 84.1 | 73.5 | 79.9 |
| Graphix-T5-3B + PICARD (Li et al., 2023b) | - | - | 81.0 | 75.0 | 77.6 |
| T5-3B + PICARD (Scholak et al., 2021) | - | - | 79.3 | 69.4 | 75.1 |
| SFT Llama2-7B (Li et al., 2024c) | 45.37 | 46.98 / - | 77.8 | 73.0 | - |
| SENSE-7B (Yang et al., 2024) | 51.8 | - | 83.2 | 81.7 | 83.5 |
| SFT CodeS-1B (Li et al., 2024c) | 49.54 | 51.07 / 62.49 | 77.8 | 71.2 | 77.5 |
| SFT CodeS-3B (Li et al., 2024c) | 55.02 | 56.54 / 70.96 | 82.2 | 76.3 | 81.9 |
| SFT CodeS-7B (Li et al., 2024c) | 57.00 | 58.80 / 72.54 | 84.7 | 79.4 | 83.3 |
| Generator-1B | 51.76 (+2.22) | 67.96 (+5.47) | 80.2 (+2.4) | 73.7 (+2.5) | 77.0 (-0.5) |
| READ-SQL-1B | 52.87 (+3.33) | 69.04 (+6.55) | 80.7 (+2.9) | 74.3 (+3.1) | 78.8 (+1.3) |
| Generator-3B | 56.98 (+1.96) | 72.43 (+1.80) | 83.8 (+1.6) | 77.7 (+1.4) | 80.9 (-1.0) |
| READ-SQL-3B | **57.37 (+2.35)** | **72.76 (+1.80)** | 84.2 (+2.0) | 78.2 (+1.9) | 81.2 (-0.7) |

outperforms SFT CodeS with the same model size, even showing a 0.22% improvement in VES with READ-SQL-3B over SFT CodeS-7B. Overall, READ-SQL sets a new state-of-the-art (SOTA) performance for models of the same size. (2) On the Spider benchmark, READ-SQL also consistently surpasses SFT CodeS with the same model size, though the improvement is less significant, and READ-SQL-3B does not outperform SFT CodeS-7B. We hypothesize that this is due to the data distribution in Spider, which limits the impact of our data augmentation. A detailed analysis of the augmented data is provided in Appendix A.3. Moreover, further analysis can be found in Sec. 4.3.

Table 2: Evaluation of READ-SQL on Spider variants: Values in parentheses record READ-SQL improvement over SFT CodeS with the same model size.

| Methods | Spider-Syn | | Spider-Realistic | | Spider-DK |
|---|---|---|---|---|---|
| | EX (%) | TS (%) | EX (%) | TS (%) | EX (%) |
| SQL-PaLM+PaLM 2 (Sun et al., 2024) | 74.6 | - | 77.6 | - | 66.5 |
| FastRAT$_{ext}$+GPT-4 (Shen et al., 2024) | 74.4 | - | 80.9 | - | 72.3 |
| TA-SQL+GPT-4 (Qu et al., 2024) | - | - | 79.5 | - | 72.9 |
| DART-SQL+GPT-3.5 (Mao et al., 2024) | - | - | 79.3 | - | 71.4 |
| ChatGPT (Li et al., 2023c) | 58.6 | 48.5 | 63.4 | 49.2 | 62.6 |
| RESDSQL-3B + NatSQL (Li et al., 2023a) | **76.9** | 66.8 | 81.9 | 70.1 | 66.0 |
| T5-3B + PICARD (Scholak et al., 2021) | 69.8 | 61.8 | 71.4 | 61.7 | 62.5 |
| SENSE-7B (Yang et al., 2024) | 72.6 | 64.9 | **82.7** | 75.6 | **77.9** |
| SFT CodeS-1B (Li et al., 2024c) | 64.7 | 56.9 | 70.1 | 62.0 | 63.2 |
| SFT CodeS-3B (Li et al., 2024c) | 73.1 | 65.4 | 78.9 | 72.8 | 70.3 |
| SFT CodeS-7B (Li et al., 2024c) | 74.8 | **67.4** | 82.3 | **76.8** | 72.9 |
| Generator-1B | 65.1 (+0.4) | 57.3 (+0.4) | 72.2 (+2.1) | 62.8 (+0.8) | 65.8 (+2.6) |
| READ-SQL-1B | 65.0 (+0.3) | 57.1 (+0.2) | 71.5 (+1.6) | 60.8 (-1.0) | 65.6 (+2.4) |
| Generator-3B | 73.9 (+0.8) | 66.6 (+1.2) | 81.1 (+2.1) | 72.8 (+0.0) | 69.9 (-0.4) |
| READ-SQL-3B | 74.0 (+0.9) | 66.4 (+1.0) | 80.3 (+1.4) | 73.2 (+0.4) | 71.6 (+1.3) |

### 4.3 EVALUATION ON ROBUSTNESS BENCHMARKS

Table 2 reports the robustness of READ-SQL across three Spider variants: Spider-Syn, Spider-Realistic, and Spider-DK, highlighting the following: (1) In three Spider variants, READ-SQL out-performs SFT CodeS of the same model size, demonstrating the advantage of READ-SQL, but in the TS indicator in Spider-Realistic, READ-SQL lags slightly behind. (2) In Spider-DK, READ-SQL shows better performance at 1B than SFT CodeS, but the performance improvement is smaller in 3B. (3) After adding Corrector, the overall level will be slightly lowered. Upon analyzing the data, we observe that Spider contains less information for rows and columns than BIRD, which provides less benefit, or even harm, for the self-correction in READ-SQL's Corrector. We compared the performance of Generator and SFT CodeS at different difficulties, as shown in the Appendix A.5.9.

### 4.4 ABLATION STUDIES

**Effect of Key Components**  Table 3 reports the ablation studies of READ-SQL-3B on BIRD's Dev, highlighting the following: (1) Removing "new pairs" (i.e., augmented (question, SQL) pairs) during Generator training results in a 0.53% drop in EX; (2) Removing "reason pairs" (,i.e., (question, reason) pairs) causes a more significant drop in EX by 1.18%; (3) Removing the Corrector yields a slight drop in EX by 0.39%. These ablation studies demonstrate the critical roles of all three components in

Table 3: Ablation studies in READ-SQL

|  | EX (%) | VES (%) |
| --- | --- | --- |
| READ-SQL-3B | 57.37 | 72.76 |
| -w/o new pairs | 56.84 (-0.53) | 72.41 (-0.35) |
| -w/o reason pairs | 56.19 (-1.18) | 71.29 (-1.47) |
| -w/o Corrector | 56.98 (-0.39) | 72.46 (-0.30) |

READ-SQL, with reason being especially important in offering reasoning information for generating SQLs while bridging the gap between questions and SQLs.

Table 4: Extensibility studies of READ-SQL's Corrector

|  | EX (%) | VES (%) | API tokens (avg) |
| --- | --- | --- | --- |
| DeepSeek-Coder-1.3B | 48.57 | 63.39 | - |
| DeepSeek-Coder-1.3B + Corrector-3B | **50.52** (+1.95) | **65.49** (+2.1) | - |
| Granite-3B-Code | 52.93 | 68.33 | - |
| Granite-3B-Code + Corrector | **53.39** (+0.46) | **68.47** (+0.08) | - |
| SFT CodeS-3B (Li et al., 2024c) | 55.02 | 70.96 | - |
| SFT CodeS-3B + Corrector-3B | **55.48** (+0.46) | **71.08** (+0.14) | - |
| READ-SQL-3B w/o Corrector | 56.98 | 72.46 | - |
| READ-SQL-3B | 57.37 (+0.39) | 72.76 (+0.3) | - |
| Generator-3B + DIN-SQL's Self-Correction (GPT-4o) | 55.93 (-1.05) | 71.28 (-1.18) | 4501 (only input) |
| Generator + Standard Self-Correction (GPT-4o) | 60.69 (+3.71) | 77.64 (+5.18) | **2039** |
| Generator + Corrector (GPT-4o) | **61.21** (+4.23) | **78.01** (+5.56) | 2619 |

**Extensibility Studies of Corrector**  Table 4 reports the results of deploying READ-SQL's Corrector in various base models, tested on BIRD Dev set, with both the Generator and the Corrector in READ-SQL having a model size of 3B. We use the same training method and training data as Generator to fine-tune DeepSeek-Coder-1.3B (Guo et al., 2024) and Granite-3B-Code (Mishra et al., 2024). The results highlight: (1) When integrating READ-SQL's Corrector into the base models, performance improves accordingly. Notably, READ-SQL without the Corrector still outperforms SFT CodeS-3B, even after SFT CodeS-3B deploys the Corrector. (2) The last two rows show that the prepared data for the Corrector can be utilized with an LLM, such as GPT-4o, to boost performance by adding only a relatively small number of tokens (around 580). The prompts for standard self-correction and for using our Corrector's data with GPT-4o are provided in Appendix A.2.

**Impact of Beam Size and Training Methods**  Figure 5 shows the impact of the beam size in READ-SQL's Generator's outputs and various training methods on BIRD Dev. Usually, larger beam sizes yield more low-confidence clauses and result in retrieving more supplemental items. Figure 5a shows that READ-SQL yields the best performance when the size is 4 and 6 for READ-SQL-1B and READ-SQL-3B, respectively. Figure 5b shows that: (1) Employing the same training data as

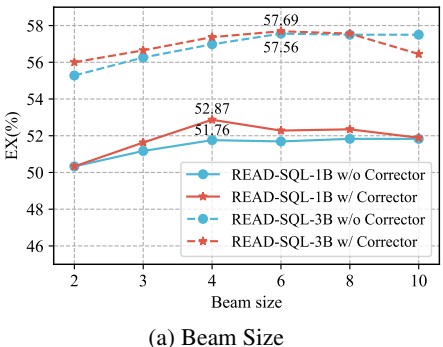 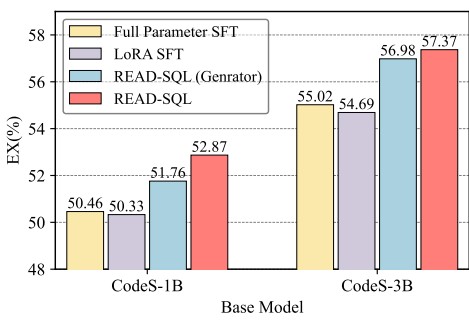

(a) Beam Size           (b) Training Methods

Figure 5: The impact of beam size and training methods

Table 5: Comparison of READ-SQL-3B and SFT CodeS-3B on three types of errors

|  | unwanted constraints | misinterpreted constraints | omitted constraints |
|---|---|---|---|
| SFT CodeS-3B ✗, READ-SQL-3B✓ | 25 | 52 | 41 |
| READ-SQL-3B ✗, SFT CodeS-3B✓ | 18(↓ 28%) | 43(↓ 17.3%) | 21(↓ 48.7%) |

SFT CodeS (Li et al., 2024c), LoRA-supervised fine-tuning (SFT) performs slightly worse than full-parameter SFT; (2) The Generator in READ-SQL, trained via the multi-task framework on top of LoRA, achieves better performance than full-parameter SFT while also improving training efficiency; (3) Additionally, the Corrector in READ-SQL further enhances performance, achieving the best results in both tests. Finally, we show the running analysis comparison of READ-SQL and SFT CodeS in the Appendix A.5.5.

**Performance of READ-SQL in three typical types of errors**  To evaluate the effectiveness of READ-SQL in addressing the three types of errors shown in Figure 1, we compared READ-SQL-3B with SFT CodeS-3B using the BIRD Dev. We present the frequency of these error types in two scenarios in Table 5: cases where READ-SQL was correct but SFT CodeS-3B was wrong, and vice versa. The results show that READ-SQL significantly reduced the occurrence of all three error types, particularly omission errors, which decreased by 48.7%. Additionally, we provide examples in the Appendix A.5.9 where READ-SQL is correct and SFT CodeS is incorrect across the three error types.

## 5 CONCLUSION

We propose READ-SQL, a novel framework that leverages READER, a key module to parse SQLs into clauses, sub-SQLs, and reasoning paths to enhance text-to-SQL tasks. READ-SQL consists of two main models: the Generator and the Corrector. The Generator is trained on both original and augmented data to recognize subtle differences between questions and SQLs while improving reasoning capabilities. Additionally, the Corrector applies READER in post-processing to ensure precise self-correction. Experimental results demonstrate that READ-SQL significantly improves strong baselines of the same model size, setting a new SOTA. Furthermore, the Corrector can be deployed on any base model, and the post-processed SQL generated by the Generator can be fed into an LLM to enhance text-to-SQL performance, underscoring its wide applicability.

Several future directions are promising: (1) While READER effectively parses SQLs, this work only utilizes partial information. Further exploration of READER's parsed information could improve performance. (2) The Corrector's effectiveness may be limited when supplemental items are unreliable due to insufficient schema information, suggesting the need for more robust methods to identify those low-confidence items. (3) Due to computational constraints, we conduct experiments on a model up to 3B. Scaling up both the model and data presents an interesting opportunity to unlock the full potential of READ-SQL.

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

# A  APPENDIX

## A.1  DETAILS OF READER

---

**Algorithm 1:** READER

---

    **Input:** $SQL$
    **Output:** $ConstraintIds, sub-SQLs, ReasoningPaths$
1   $sub-SQLs \leftarrow set(\ ), ReasoningPaths \leftarrow [\ ]$;
2   $ast \leftarrow$ ParserToAst $(SQL)$                     ▷ Parse SQL into its AST
3   $ConstraintIds \leftarrow$ GetConstraints $(ast)$         ▷ Get all constraints into a list
4   $BinaryAst \leftarrow$ AstNode $(ast)$              ▷ Encapsulate as binary tree
5   **for** $id$ in $ConstraintIds$ **do**
6      TravrseAST $(BinaryAst, id, sub-SQLs)$    ▷ Traverse each constraint to construct a new ast

7   **Function** TravrseAST $(node, id, sub\text{-}SQLs)$ **:**
8      $sub-SQLs.add(node.ast.tosql(\ ))$         ▷ Parse the current node's AST into SQL
9      **if** not $node.left$ **and** not $node.right$ **then**
10         $ast =$ DeleteConstraint $(node.ast, id)$      ▷ Delete the current constraint from the AST
11         $node.left \leftarrow$ AstNode $(ast)$    ▷ Set the new AST as the left child of the current node
12         $node.right \leftarrow$ AstNode $(node.ast)$        ▷ Keep the original AST as the right child
13         **return** 0
14      TravrseAST $(node.left, id, sub-SQLs)$      ▷ Pre-order traversal, visit child nodes after root
15      TravrseAST $(node.right, id, sub-SQLs)$;
16      **return** 0

17   $Leaves \leftarrow$ GetAllLeaves $(BinaryAst)$      ▷ Obtain all the leaf nodes of the perfect binary tree
18   $ReasoningPaths \leftarrow$ CombinePath $(Leaves)$    ▷ Construct inference paths between the leaf nodes
19   **return** $ConstraintIds, sub-SQLs, ReasoningPaths$

---

### A.1.1  EXPLANATION OF ALGO. 1

Algorithm 1 outlines the procedure of READER:

- Line 2 runs a parser to parse the SQL into an AST.
- Line 3 gets the set of constraints in the AST based on pre-defined rules. Detailed information about these rules is provided in the Appendix A.1.2.
- Line 4 initializes a binary tree to store intermediate results, with the AST placed in the root node.
- Lines 5-6 deletes each constraint and save the result in a binary tree.
- Lines 7-16 defines a method to remove constraint in the AST from the leaf nodes of the binary tree, assigning the resulting new AST and a backup of the original AST as the child nodes.
- Lines 17 extract all possible sub-SQLs from the leaf nodes of the binary tree.
- Lines 18 employ a bottom-up breadth-first approach to identify the inclusion relationships of constraints in sub-SQLs and retrieve all existing reasoning paths.

Hence, given a question, "What is the name of a person who is 18 years old?" and its ground-truth SQL, "SELECT name FROM person WHERE age = 18", we can obtain the corresponding results as follows:

- In line 3 of Algorithm. 1 or Step (2) in Figure 3, identify three clauses: "SELECT name", "FROM person", and "WHERE age = 18".
- In 4-16 of Algorithm. 1 or Step (3) in Figure 3: remove constraints to obtain sub-SQLs. First, the method deletes "SELECT name" to obtain "SELECT * FROM person WHERE age = 18", along with a backup of the original SQL. Next, the method removes "WHERE age = 18" from both results, yielding "SELECT * FROM person" and "SELECT name FROM person", thereby generating all sub-SQLs.
- In 17-18 of Algorithm. 1 or Step (4) in Figure 3: using "SELECT * FROM person" as the starting point of the reasoning path, the next points can be "SELECT name FROM person" and "SELECT * FROM person WHERE age = 18". The endpoints of both reasoning paths represent the ground-truth SQL.

It is important to note that this case can be extended to more complicated examples, as illustrated in Figure 16. In the related example, "How many movies directed by Francis Ford Coppola have a

Table 6: Operational type and non-operational type nodes

| Non-operational type nodes | Column, Table, Identifier, Literal, Null, Datatype, TableAlias |
|---|---|
| Operational type nodes | **Main body types:** 
 SELECT , FROM , WHERE , EXISTS , IIF , CASE , CASE WHEN , JOIN , INNER JOIN , BETWEEN , LIKE , LIMIT , ORDER BY , GROUP BY , DESC , ASC , HAVING , SUBQUERY , WINDOW , OVER 
 **Arithmetic operation types**: 
 AND , OR , ADD (+), SUB (-) , MUL (*) , DIV (/) , GT (>) , GTE (>=) , LT (<) , LTE (<=), EQ (=), NEQ (!=) , UNION , INTERSECT 
 **Built-in function types**: 
 AVG , COUNT , MAX , MIN , ROUND , SUM , ABS , NOW , CAST |

popularity of more than 1,000? Indicate the highest number of likes that each critic has received per movie, if applicable", four additional clauses will be included in line 3 of Algorithm. 1 or Step (2) in Figure 3: "SELECT count(movies.movie_title)", "SELECT ratings.critic", "INNER JOIN movies ON ratings.movie_id = movies.movie_id", "WHERE movies.director_name = 'Francis Ford Coppola'" and "WHERE movies.movie_popularity > 1000". We can further derive the corresponding sub-SQLs and the reasoning paths accordingly.

### A.1.2 NODE TYPES

For each node in AST, they can be categorized into the following types (see details in Table 6):

- **Non-operational type nodes** include columns, tables, identifiers, Literals, etc.
- **Operational types nodes** are defined as nodes whose operation objects are non-operational nodes. In simple terms, an operation node can be defined as a constraint.

Generally speaking, the subquery node in SQL represents a distinct SQL query. We treat it as a separate constraint and perform constraint decomposition on the subquery independently. A subquery (also known as an inner query or nested query) is a query that is embedded within another SQL query, as shown in Figure 8, Sub-query 1 is a subsquery node in sub-query 2.

For each sub-SQL, we can determine the constraints based on the following criteria:

- Dependencies exist between constraints, necessitating careful judgment before deletion. For example, the constraint "WHERE movies.movie_popularity > 1000" relies on the JOIN constraint "INNER JOIN movies ON ratings.movie_id = movies.movie_id." This indicates that the column referenced in the WHERE clause belongs to the movies table. Consequently, when deleting the JOIN constraint, the corresponding WHERE constraint must also be removed; otherwise, the JOIN constraint cannot be deleted. Nodes with dependencies include: GROUP BY and HAVING; and JOIN nodes along with all constraints involving the tables in the JOIN.
- Constraints can be merged to reduce the number of generated sub-SQLs. For instance, the constraints "SELECT name" and "SELECT year" can be combined because both columns belong to the person table. However, if the columns originate from different tables, they cannot be merged. The only node types that can be merged are non-operation column node types.

### A.1.3 READER RESULTS

We apply this READER to BIRD and Spider datasets. For the training set, we limit the number of sub-SQLs to 256, and do not perform data enhancement on samples that are too complex. We count the corresponding results in each dataset, as shown in the Table 7. It can be found that each sample in the BIRD dataset can be decomposed into 14.4 sub-SQLs, the average number of reasoning paths is 199.7, and the number of sub-SQLs involved in each reasoning path is 4.8. For the Spider dataset, the structure of SQL is simpler, so the sub-SQLs and reasoning paths generated by the algorithm are less than those of the BIRD dataset.

Table 7: Results Statistics after READER Processing of the BIRD and Spider Datasets

| Dataset | Number of Sub-SQLs | | | Number of Reasoning Paths | | | Length of Reasoning Path | | |
|---|---|---|---|---|---|---|---|---|---|
| | Avg | Max | Min | Avg | Max | Min | Avg | Max | Min |
| BIRD Train Set | 14.4 | 256 | 1 | 199.7 | 130,704 | 0 | 4.8 | 11 | 2 |
| BIRD Dev Set | 13.7 | 192 | 2 | 97.6 | 20,160 | 1 | 4.8 | 11 | 2 |
| Spider Train Set | 8.2 | 240 | 2 | 42.4 | 99,360 | 1 | 3.9 | 11 | 2 |
| Spider dev Set | 8.0 | 50 | 2 | 12.3 | 560 | 1 | 3.9 | 8 | 2 |

Table 8: Comparison of READER and existing SQL decomposers

| | Split Clause | Synthetic sub-SQL | multiple paths | Support for complex syntax |
|---|---|---|---|---|
| READER | ✓(Refinement) | ✓ | ✓ | ✓ |
| STEPS | ✓ | × | × | ✓ |
| DeSQL | ✓(Refinement) | ✓ | × | × |

### A.1.4 COMPARISON OF READER AND EXISTING SQL DECOMPOSERS

READER is a self-developed method based on SQLglot. Compared with the existing SQL decomposition, it can divide SQL clauses more finely, split all possible executable sub-SQLs, and obtain multiple reasoning paths. We compared it with two recently published works, such as STEPS (Tian et al., 2023) and DeSQL (Haroon et al., 2024).

- Neither STEPS nor DeSQL can support outputting multiple reasoning paths.
- STEPS cannot fine-grain SQL, resulting in clauses not being independent units.
- DeSQL has limited SQL structures involved, cannot handle nested queries, etc., and can only decompose simple SQL.

We show examples of the three methods in Table 9 and Table 10.

### A.2 PROMPT TEMPLATES

In our work, there are totally four kinds of prompts:

- **Sub-question generation prompt**: This prompt is designed to generate augmented (sub-question, sub-SQL) pairs based on the provided sub-SQLs; refer to the prompt in Fig. 9 and an example of a (sub-question, sub-SQL) pair in Figure 10.
- **Reason generation prompt**: This prompt generates the reason or description of a chain of thought (CoT) based on the given reasoning path; see the prompt in Fig. 11 and an example of a (question, reason) pair in Figure 12.
- **Self-Correction prompt**: This prompt replaces the Corrector in READ-SQL with a large language model (LLM), such as GPT-4; see the prompt in Figure 14. The goal is to enable the model to reevaluate low-confidence constraints using the additional information provided.
- **Standard Self-Correction prompt**: This prompt provides only the beam search results and the original database schema information, excluding any additional details retrieved from low-confidence constraints and basic SQL. See the prompt in Figure 15.
- **DIN-SQL's Self-Correctio prompt**: We have not modified the self-correction prompt for DIN-SQL. The input consists of the schema, related row displays with corresponding column descriptions, the question, a hint (evidence), and the SQL query requiring correction. DIN-SQL employs a chain-of-thought (CoT) approach, prompting the model to first generate reasoning steps before producing the revised SQL query. See the prompt in Figure 13.

### A.3 DETAILS OF DATA AUGMENTATION

As shown in Table 7, on average, READER parses over 10 sub-SQLs and more than 90 reasoning paths in the BIRD dataset, while in the Spider dataset, it parses over 8 sub-SQLs and more than 10 reasoning paths. Given the large quantities, we need a strategy for selecting them to optimize costs. All data enhancements rely on the training set. Firstly, for sub-SQLs, we only select those with

Table 9: Decomposition results of three SQL decomposers under complex syntax

| SQL query | SELECT MAX(horsepower) - (SELECT MAX (horsepower) FROM cars_data A JOIN car_names B ON A.id=B.makeid WHERE B.model='fiat') AS diff FROM cars_data A JOIN car_names B ON A.id=B.makeid WHERE B.model='bmw' |
|---|---|
| READER | subquey 1: 'Select': ['MAX(horsepower)'], 'Where': ["car_names.model = 'fiat'"], 'Table': ['FROM cars_data', 'JOIN car_names AS B'] subquey 2: 'Select': ['MAX(horsepower)', "(subquery 1)"], 'Where': ["car_names.model = 'bmw'"], 'Table': ['FROM cars_data', 'JOIN car_names AS B'] |
| STEPS | subquey 1: [ SELECT MAX ( horsepower ), FROM cars_data A JOIN car_names B ON A.id = B.makeid, WHERE B.model = "fiat"] subquery2: [SELECT MAX ( horsepower ) - ( subquery 1), FROM cars_data A JOIN car_names B ON A.id = B.makeid WHERE B.model = fiat) AS diff, WHERE B.model = "bmw", ] |
| DeSQL | Unable to disassemble, nested query syntax is not supported |

Table 10: Results of three SQL parsers generating sub-SQLs

| SQL query | SELECT MAX (horsepower) FROM cars_data A JOIN car_names B ON A.id=B.makeid WHERE B.model='fiat' |
|---|---|
| READER | sub-SQLs: 1: 'SELECT * FROM cars_data AS A', 2: 'SELECT * FROM cars_data AS A JOIN car_names AS B ON A.id = B.makeid', 3: 'SELECT MAX(horsepower) FROM cars_data AS A', 4: "SELECT * FROM cars_data AS A JOIN car_names AS B ON A.id = B.makeid WHERE B.model = 'fiat'", 5: 'SELECT MAX(horsepower) FROM cars_data AS A JOIN car_names AS B ON A.id = B.makeid', 6: "SELECT MAX(horsepower) FROM cars_data AS A JOIN car_names AS B ON A.id = B.makeid WHERE B.model = 'fiat'" inference path: [1, 2, 4, 6] or [1, 2, 5, 6] or [1, 3, 5, 6] |
| STEPS | Unable to generate sub-SQL. |
| DeSQL | sub-SQLs: 1: 'SELECT * FROM cars_data AS A', 2: 'SELECT * FROM cars_data AS A JOIN car_names AS B ON A.id = B.makeid', 3: "SELECT * FROM cars_data AS A JOIN car_names AS B ON A.id = B.makeid WHERE B.model = 'fiat'", 4: "SELECT MAX(horsepower) FROM cars_data AS A JOIN car_names AS B ON A.id = B.makeid WHERE B.model = 'fiat'" inference path: [1,2,3,4] |

constraints that differ from the original SQL by two or fewer in the BIRD dataset. In contrast, some sub-SQLs in the Spider dataset may correspond to multiple original SQLs. This is because Spider employs a similar approach of limiting condition deletion to construct its dataset. To gather more training data, we collect sub-SQLs for Spider, selecting those with constraints that differ from the original SQL by three or fewer.

Additionally, some sub-SQLs may contain extraneous constraints, such as unnecessary JOIN operations. Removing or retaining JOIN conditions does not impact the execution results of SQL, so we do not enhance data in these cases. Finally, we manually review the quality of the generated data, resulting in 2,110 sub-question/sub-SQL pairs for the BIRD dataset and 1,108 pairs for Spider. For reasoning paths, we randomly select one reasoning path from the original question to generate descriptive information and manually eliminate low-quality data. Ultimately, we obtain 9,108 sub-question/sub-SQL pairs for the BIRD dataset and 8,505 pairs for Spider.

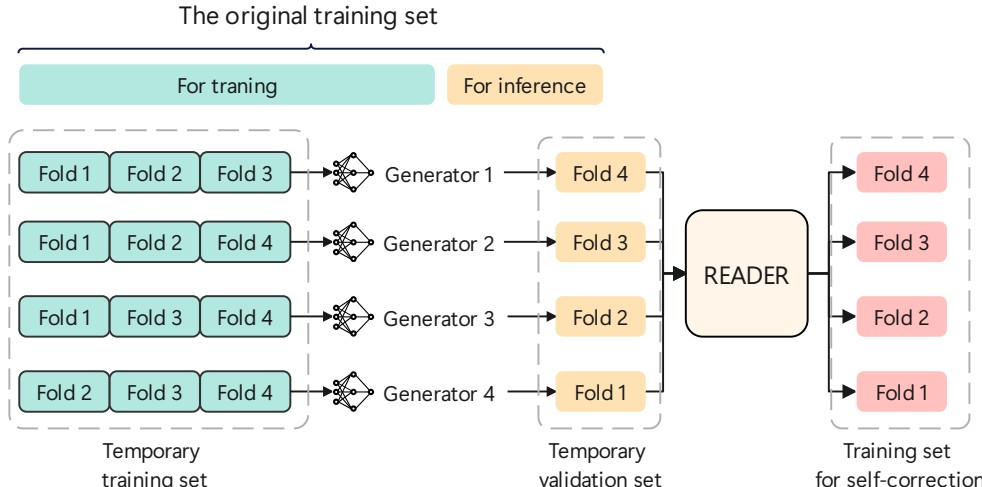

Figure 6: Constructing a training set for self-correction

## A.4 DETAILS OF SELF-CORRECTION

To obtain training data for self-correction, we simulate reasoning using the training set. In this process, we split the original training set by database into four folds, as shown in Figure 6. Each fold is designed to be as consistent as possible, ensuring that all samples from a database belong to only one fold. We use one fold as the reasoning dataset while the other three folds serve as training data. After completing four training iterations, READER parses the reasoning results and organizes the input. This approach ultimately yields a training set for self-correction.

## A.5 DETAILS OF EXPERIMENTS

### A.5.1 DETAILS OF DATASETS

In our work, we conduct experiments on the following datasets:

- **BIRD** (Li et al., 2023c) is the first cross-domain, large-scale benchmark specifically designed to bridge the gap between academic research and real-world applications in text-to-SQL parsing. It features a substantial dataset comprising 12,751 text-to-SQL pairs, 95 databases across 37 professional domains, and a total size of 33.4 GB. In comparison to Spider (Yu et al., 2018b) and WikiSQL (Zhong et al., 2017), BIRD-SQL emphasizes database content and aligns more closely with real-world scenarios. However, due to hardware limitations of the BIRD submission platform, we are unable to evaluate our model on its test set.
- **Spider** (Yu et al., 2018b) is a large-scale semantic parsing and text-to-SQL dataset annotated by 11 students from Yale University. It includes a training set with 8,659 samples, a development set with 1,034 samples, and a test set with 2,147 samples, covering 200 distinct databases across 138 domains. Of these, 160 databases are allocated for training and development, while 40 are designated for testing.
- **Spider-DK** (Gan et al., 2021b), **Spider-Syn** (Gan et al., 2021a), and **Spider-Realistic** (Deng et al., 2021) are variants of the original Spider dataset, designed to resemble queries that users might ask in real-world scenarios. These variants allow for a more comprehensive evaluation of the model's robustness in the text-to-SQL task.

Table 11 reports the statistics of Spider and BIRD, where the SQL queries in BIRD are typically more complex than those in the Spider dataset.

Table 11: Statistics of BIRD and Spider

|  | Examples | Databases | tables | Domains | Row/database |
|---|---|---|---|---|---|
| BIRD | 12,751 | 92 | 7.3 | 37 | 549k |
| Spider | 10,181 | 200 | 5.1 | 138 | 2k |

Table 12: Statistics of three typical categories of text-to-SQL errors

|  | Unwanted constraint | Misinterpreted constraint | Omitted constraint | Error gold |
|---|---|---|---|---|
| percentage | 13% | 47% | 43% | 7% |

### A.5.2 DETAILS OF THREE TYPICAL ERRORS IN TEXT-TO-SQL

We randomly select 100 error samples from the 690 total errors in SFT CodeS-3B and manually categorized them, It is important to note that a single erroneous sample may contain multiple errors, the results are in Table 12.

### A.5.3 DETAILS OF TRAINING SETTINGS

The maximum input length is set to 4,096 tokens. The learning rate scheduling strategy employs cosine decay, starting with a learning rate of $1 \times 10^{-4}$. The model is trained for 6 epochs, while the epoch count is increased to 10 during multi-task fine-tuning due to the larger training dataset. All LoRA parameters utilize the default settings from the Transformers library: the rank $r$ of the low-rank matrix is 16, the scaling factor is 32, and the dropout rate is 0.1. The LoRA module is added to the projection layers of the model. During inference, the maximum output token length is capped at 256 tokens. Greedy decoding is used with a default beam size of 4. The first executable SQL query is selected as the prediction result for both the Generator and the Corrector.

Table 13: Effect of train parameter $\lambda$

| $\lambda$ | 0.6 | 0.8 | 1.0 |
|---|---|---|---|
| EX(%) | 50.72 | 51.76 | 51.37 |

### A.5.4 EFFECT OF TRAIN PARAMETER

Table 13 presents a comparison of the training performance based on the hyperparameter $\lambda$. The data shown reflects the performance of Generator-1B on the BIRD Dev under different training parameters. The results indicate that when $\lambda$ is set to 0.8, the weighting between the question/SQL pairs and the question/Reason pairs is optimized. If the weight of the Reason pairs is too low, the model fails to capture reasoning information effectively. Conversely, if $\lambda$ is set too high, the model tends to overemphasize the reasoning pairs, neglecting the information from the question/SQL pairs. Both scenarios can diminish the model's effectiveness in generating SQL.

### A.5.5 RUNTIME ANALYSIS OF READ-SQL

We conducted a comparative analysis of READ-SQL and SFT-CodeS with respect to model size, training time, inference time, and memory usage on NVIDIA RTX 3090 GPUs with 24 GB of memory. As READ-SQL-3B slightly outperforms SFT-CodeS-7B on the BID dataset, we use SFT-CodeS-7B as a benchmark. The experiments were carried out using a two-card setup for training and a single-card setup for inference, with results presented in the Table 14.

READ-SQL offers the advantage of achieving high performance with minimal memory usage. Leveraging the LoRA fine-tuning method, it enables efficient training and inference in low-resource environments compared to full fine-tuning. Furthermore, using only the Generator-3B improves performance by 1.96% compared to a single 3B model, with almost no increase in inference time. In addition, READ-SQL-3B surpasses SFT CodeS-7B by nearly 0.37% with smaller inference time and less memory usage.

Table 14: Comparative analysis of runtime between READ-SQL and SFT CodeS. READ-SQL comprises two components: the Generator and the Corrector, with training and inference divided into two stages. Data is presented in a sequence where the Generator precedes the Corrector. As the Generator introduces additional data, the training time is slightly longer.

| | Model size | Training time | Estimated memory usage | Inference time (s/sample) | BIRD Dev(EX) |
|---|---|---|---|---|---|
| SFT CodeS-3B | 3B | out of memory | 6G | 2.81 | 55.02 |
| SFT CodeS-7B | 7B | out of memory | 14G | 6.51 | 57.00 |
| READ-SQL-3B | 3B+0.02B | 8h 23m + 3h 24m | 6.1G | 4.65(2.97+1.68) | **(56.98/57.37)** |

Table 15: Effect of data organization in fine-tuning

| | RSF-1B | Generator-1B | RSF-3B | Generator-3B |
|---|---|---|---|---|
| EX(%) | 40.94 | 51.76 | 48.63 | 56.98 |

### A.5.6 THE IMPACT OF DATA ORGANIZATION IN FINE-TUNING

We use multi-task fine-tuning to provide question/SQL pairs and question/reason pairs to the model simultaneously, guiding it to output different content through pre-specified tokens. Additionally, we experimented with the Reason-SQL Formatting (RSF) method of data organization, where the model first outputs the reason and then the SQL. The output format is: `reason:(reason text); SQL: (SQL query).` We trained on the BIRD dataset, and the results on the BIRD Dev are in the Table 15.

This approach is clearly unsuitable for small language models (SLMs) for the following reasons:

- The reason output during model inference contributes to **error accumulation**, interfering with subsequent decoding and introducing unnecessary errors.
- Beam search is used for decoding, and including the reason output reduces the available space for the SQL portion, limiting the model's ability to generate diverse SQL queries.

### A.5.7 ERROR ANALYSIS

To analyze our failures, we randomly select 50 error results from the READ-SQL-3B in the BIRD Dev and conduct a thorough analysis. We identify the causes of these errors and summarize them into two primary situations:

- The model's reasoning ability in text-to-SQL tasks is insufficient. Given the rich table schema information, the model struggles to accurately identify useful data and generate the corresponding SQL clauses.
- The model's capability for schema linking to retrieve information from a large database remains inadequate, particularly in terms of value retrieval.

The specific error cases are as follows:

- **Incorrect columns exist in the SQL query.** Despite the relevant information being provided in the context, the model fails to identify the correct columns. An example is shown in the Table 25.
- **Syntax error.** Even when the model's semantics are correct, errors may arise due to variations in SQL structure. An example is shown in the Table 26.
- **The evidence information is ignored.** This causes the model to generate incorrect SQL clauses. An example is shown in the Table 27.
- **Joining additional tables.** The model generates some unnecessary tables in the SQL query. An example is shown in the Table 28.
- **Added additional operations.** The model generates incorrect or extra aggregate functions or arithmetic expressions. An example is shown in the Table 29.
- **Incorrect format.** The date format in the predicted SQL is incorrect. Even though the context provides a correct date format example. An example is shown in the Table 30.

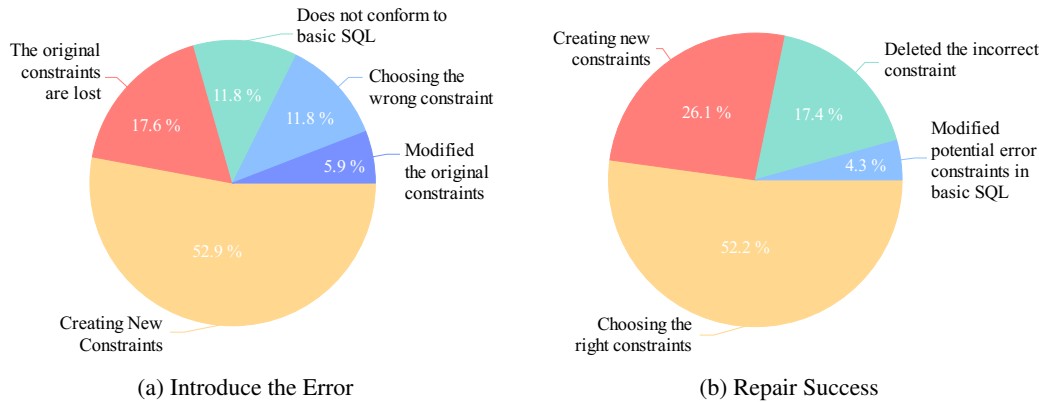

(a) Introduce the Error                    (b) Repair Success

Figure 7: Case study in Corrector

- **Value Error.** This type of error mainly occurs during the schema linking phase, when the corresponding value information is not retrieved. An example is shown in the Table 31.
- **Ignore the underlying information in the question.** The schema linking phase does not retrieve the schema related to the problem, resulting in the generation of incorrect SQL. An example is shown in the Table 32.

### A.5.8  CASE STUDY FOR CORRECTOR

To explore the internal mechanism of self-correction, we analyze the results of the Corrector-3B on the BIRD Dev. Based on the results from Generator-3B, the Corrector-3B successfully fixes 23 errors, referred to as "Repair Success." Simultaneously, it alters 17 correct SQLs, which we term "Introduce the Error."

Additionally, 856 correct SQLs remain correct after the repair, labeled as "Maintain Correctness," while 638 incorrect SQLs remain incorrect, referred to as "Repair Failure." We conduct a detailed analysis of the two scenarios: Repair Success and Introduce the Error, with the data ratios illustrated in Figure 7. We categorize the causes of these two phenomena and find that the model tends to involuntarily create new constraints, which is the primary reason for introducing errors. Additionally, the model selects the correct constraints based on the provided beam search results and the additional table schema information retrieved, which is the main factor contributing to successful repairs.

We categorize the situation of "Introduce the Error" into the following five categories:

- **Creating New Constraints:** New constraints that have not appeared in the beam search results are added, leading to errors. An example is shown in Table 33.
- **Loss of Original Constraints:** After self-correction, the constraints that should have been included are lost, resulting in incomplete SQL. An example is shown in the Table 34.
- **Non-conformity to Basic SQL:** The result after self-correction conflicts with basic SQL, as the self-correction does not build upon the original beam search results. An example is shown in the Table 35.
- **Choosing the Wrong Constraint:** During the self-correction process, an incorrect constraint is selected. An example is shown in the Table 36.
- **Adding Additional Error Constraints:** Compared to the original prediction results, self-correction introduces incorrect constraints in the beam search. An example is shown in the Table 37.

We categorize the situation of "Repair Success" into four categories:

- **Choosing the Right Constraints:** After self-correction, the Corrector selects the correct constraints from the beam search results. An example is shown in the Table 38.
- **Creating New Constraints:** Additional constraints that have not appeared in the beam search results are added, addressing previously unconsidered situations. An example is shown in the Table 39.

Table 16: Execution Accuracy (EX) across queries of varying levels of difficulty on BIRD Dev

| Model | Simple | Moderate | Challenging | Total |
|---|---|---|---|---|
| SFT CodeS-1B | 57.95 | 37.42 | 34.72 | 49.54 |
| Generator-1B | 59.89(+1.94) | 41.29(+3.87) | 33.33(-1.39) | 51.76(+2.22) |
| READ-SQL-1B | 60.56(+2.61) | 43.87(+6.45) | 31.94(-2.78) | 52.87(+3.33) |
| SFT CodeS-3B | 63.35 | 44.30 | 36.11 | 55.02 |
| Generator-3B | 65.08(+1.73) | 46.67(+2.37) | 38.19(+2.08) | 56.98(+1.96) |
| READ-SQL-3B | 65.41(+2.06) | 47.53(+3.23) | 37.50(+1.39) | 57.37(+2.35) |
| SFT CodeS-7B | 64.60 | 46.90 | 40.30 | 57.00 |

Table 17: Execution Accuracy (EX) across queries of varying levels of difficulty on Spider Dev

| Model | Easy | Medium | Hard | Extra | All |
|---|---|---|---|---|---|
| SFT CodeS-1B | 91.10 | 83.60 | 68.40 | 51.80 | 77.80 |
| Generator-1B | 91.90(+0.80) | 83.60(+0.00) | 73.60(+5.20) | 53.00(+1.20) | 80.20(+2.40) |
| READ-SQL-1B | 91.90(+0.80) | 86.80(+3.20) | 74.70(+6.30) | 53.60(+1.80) | 80.70(+2.90) |
| SFT CodeS-3B | 93.50 | 87.00 | 73.60 | 61.40 | 82.20 |
| Generator-3B | 94.80(+1.30) | 88.60(+1.60) | 79.90(+6.30) | 58.40(-3.00) | 83.80(+1.60) |
| READ-SQL-3B | 94.80(+1.30) | 89.20(+2.20) | 81.00(+7.40) | 58.40(-3.00) | 84.20(+2.00) |

- **Deleting Incorrect Constraints:** Compared to the original prediction results, the Corrector removes incorrect constraints. An example is shown in the Table 40.
- **Modifying Potential Error Constraints in Basic SQL:** The Corrector modifies potential errors within basic SQL. An example is shown in the Table 41.

### A.5.9 CASE STUDY COMPARING READ-SQL AND SFT CODES

We conduct a statistical analysis of the BIRD Dev, SPIDER Dev and three Spider variants datasets based on difficulty level.

Table 16 presents the specific results from the BIRD Dev, followed by a discussion of our conclusions:

- The Generator generally outperforms SFT-CodeS, showing the greatest improvement at moderate difficulty levels.
- With the introduction of the Corrector, there are improvements at both simple and moderate difficulty levels, though its effectiveness may decrease at more challenging difficulty levels.

Table 17 presents the specific results from the Spider Dev, followed by a discussion of our conclusions:

- The Generator exhibits the highest growth rate at hard difficulty.
- The addition of the Corrector does not lead to significant improvements.

Table 18 presents the specific results from the three Spider variants, followed by a discussion of our conclusions:

- On easy and extra-hard levels, its performance is occasionally weaker than SFT CodeS.
- On easy and extra-hard levels, its performance is occasionally weaker than SFT CodeS.
- A significant performance gap between READ-SQL and SFT CodeS is observed in Spider-DK.
- These results indicate that adding extra training data significantly benefits the model on medium-difficulty tasks but may slightly hinder performance on simpler or extremely complex tasks.
- This further suggests that READ-SQL leverages additional training data more effectively on challenging datasets, such as BIRD, compared to simpler ones like Spider.

In addition, we present examples where READ-SQL produces correct outputs while SFT CodeS generates incorrect results under three types of errors, as shown in Tables 19 to Table 24.

Table 18: Execution Accuracy (EX) on queries of varying difficulty across three Spider variants

| Dataset | Model | Easy | Medium | Hard | Extra | All |
|---------|-------|------|--------|------|-------|-----|
| Spider-Syn | SFT CodeS-1B | 80.2 | 68.0 | 58.2 | 40.2 | 64.7 |
| Spider-Syn | Generator-1B | 77.4(-2.8) | 69.5(+1.5) | 61.0(+2.8) | 39.6(-0.6) | 65.1(+0.4) |
| Spider-Syn | SFT CodeS-3B | 83.4 | 74.8 | 67.8 | 58.0 | 73.1 |
| Spider-Syn | Generator-3B | 84.3(+0.9) | 76.1(+1.3) | 70.6(+2.8) | 56.2(-1.8) | 73.9(+0.8) |
| spider-Realistic | SFT CodeS-1B | 88.1 | 77.3 | 63.6 | 41.2 | 70.1 |
| spider-Realistic | Generator-1B | 87.2(-0.9) | 79.8(+2.5) | 69.7(+6.1) | 42.3(+1.1) | 72.2(+2.1) |
| spider-Realistic | SFT CodeS-3B | 95.4 | 84.2 | 70.7 | 57.7 | 78.9 |
| spider-Realistic | Generator-3B | 93.6(-1.8) | 86.2(+2.0) | 74.7(+4.0) | 62.9(+5.2) | 81.1(+2.2) |
| spider-DK | SFT CodeS-1B | 79.1 | 69.5 | 51.4 | 40.0 | 63.2 |
| spider-DK | Generator-1B | 79.1(+0.0) | 70.7(+1.2) | 51.4(+0.0) | 50.5(+10.5) | 65.8(+2.6) |
| spider-DK | SFT CodeS-3B | 82.7 | 76.8 | 55.4 | 54.3 | 70.7 |
| spider-DK | Generator-3B | 82.7(+0.0) | 76.4(-0.4) | 62.2(+6.8) | 46.7(-7.6) | 69.9(-0.8) |

Table 19: Unwanted constraints: case 1

| | |
|---|---|
| Question | Please list top three unknown power cards that have promotional types for arena league in alphabetical order. |
| Evidence | unknown power cards refers to power is null or power = '*'; promotional types for arena league refers to promotypes = 'arenaleague'; |
| READ-SQL result | SELECT name FROM cards WHERE (power IS NULL OR power = '*') AND promotypes = 'arenaleague' ORDER BY name LIMIT 3 |
| SFT CodeS result | SELECT name FROM cards WHERE power IS NULL OR power = '*' AND promotypes = 'arenaleague' ORDER BY name DESC LIMIT 3 |
| Cause analysis | Added additional sorting constraints |

Table 20: Unwanted constraints: case 2

| | |
|---|---|
| Question | List the elements of all the triple bonds. |
| Evidence | triple bond refers to bond_type = '#'; element = 'cl' means Chlorine; element = 'c' means Carbon; element = 'h' means Hydrogen; |
| READ-SQL result | SELECT atom.element FROM connected INNER JOIN bond ON connected.bond_id = bond.bond_id INNER JOIN atom ON connected.atom_id = atom.atom_id WHERE bond.bond_type = '#' |
| SFT CodeS result | SELECT DISTINCT atom.element FROM bond INNER JOIN atom ON bond.molecule_id = atom.molecule_id WHERE bond.bond_type = '#' AND atom.element IN ('cl', 'c', 'h', 'o', 's', 'n', 'p', 'na', 'br', 'f', 'i', 'sn') |
| Cause analysis | element adds additional values, resulting in an expanded range |

Table 21: Misinterpreted constraints: case 1

| | |
|---|---|
| Question | Please list the CodeS of the schools with a total enrollment of over 500. |
| Evidence | Total enrollment can be represented by 'enrollment (k-12)' + 'enrollment (ages 5-17)'; |
| READ-SQL result | SELECT DISTINCT cdscode FROM frpm WHERE cast('enrollment (k-12)' + 'enrollment (ages 5-17)' AS REAL) > 500 |
| SFT CodeS result | SELECT 'school code' FROM frpm WHERE 'enrollment (k-12)' + 'enrollment (ages 5-17)' > 500 |
| Cause analysis | Confusing school code and cdscode |

Table 22: Misinterpreted constraints: case 2

| | |
|---|---|
| Question | Sort in descending order all patients by birthday for male patient with albumin not within range. |
| Evidence | albumin not within range refers to alb <= 3.5 or alb >= 5.5; male = sex = 'M'; |
| READ-SQL result | SELECT patient.id FROM patient INNER JOIN laboratory ON patient.id = laboratory.id WHERE patient.sex = 'M' AND (laboratory.alb <= 3.5 OR laboratory.alb >= 5.5) ORDER BY patient.birthday DESC |
| SFT CodeS result | SELECT patient.id FROM patient INNER JOIN laboratory ON patient.id = laboratory.id WHERE patient.sex = 'M' AND (laboratory.alb < 3.5 OR laboratory.alb > 5.5) ORDER BY patient.birthday DESC |
| Cause analysis | Symbols mixed up |

Table 23: Omitted constraints: case 1

| | |
|---|---|
| Question | List the patient ID, sex and birthday who has abnormal white blood cell count. Group them by sex and list the patient by age in ascending order. |
| Evidence | abnormal white blood cell count refers to wbc <= 3.5 or wbc >= 9.0; |
| READ-SQL result | SELECT patient.id, patient.sex, patient.birthday FROM patient INNER JOIN laboratory ON patient.id = laboratory.id WHERE laboratory.wbc <= 3.5 OR laboratory.wbc >= 9.0 GROUP BY patient.id, patient.sex, patient.birthday ORDER BY patient.birthday ASC |
| SFT CodeS result | SELECT patient.id, patient.sex, patient.birthday FROM patient INNER JOIN laboratory ON patient.id = laboratory.id WHERE laboratory.wbc BETWEEN 3.5 AND 9.0 |
| Cause analysis | Lost the ORDER by condition. |

Table 24: Omitted constraints: case 2

| | |
|---|---|
| Question | Between San Diego and Santa Barbara, which county offers the most number of schools that does not offer physical building? Indicate the amount. |
| Evidence | Does not offer physical building' means virtual = F in the database. |
| READ-SQL result | SELECT county, count(school) FROM schools WHERE county IN ('San Diego', 'Santa Barbara') AND virtual = 'F' GROUP BY county ORDER BY count(school) DESC LIMIT 1 |
| SFT CodeS result | SELECT county, count(cdscode) FROM schools WHERE county IN ('San Diego', 'Santa Barbara') AND virtual = 'F' GROUP BY county |
| Cause analysis | There is no guarantee of the required number of rows. |

Table 25: Incorrect columns exist in the SQL query

**Question:** Please list the lowest three eligible free rates for students aged 5-17 in continuation schools.
**Evidence:** Eligible free rates for students aged 5-17 = 'free meal count (ages 5-17)' / 'enrollment (ages 5-17)';

**Predicted SQL:** SELECT cast('free meal count (ages 5-17)' AS REAL) / 'enrollment (ages 5-17)' FROM frpm INNER JOIN schools ON frpm.cdscode = schools.cdscode WHERE schools.edopsname = 'Continuation School' ORDER BY cast('free meal count (ages 5-17)' AS REAL) / 'enrollment (ages 5-17)' LIMIT 3

**Gold SQL:** SELECT 'Free Meal Count (Ages 5-17)' / 'Enrollment (Ages 5-17)' FROM frpm WHERE 'Educational Option Type' = 'Continuation School' AND 'Free Meal Count (Ages 5-17)' / 'Enrollment (Ages 5-17)' IS NOT NULL ORDER BY 'Free Meal Count (Ages 5-17)' / 'Enrollment (Ages 5-17)' ASC LIMIT 3

Table 26: Syntax error

**Question:** For the school with the highest average score in Reading in the SAT test, what is its FRPM count for students aged 5-17?
**Evidence:** None

**Predicted SQL:** SELECT frpm.'frpm count (ages 5-17)' FROM satscores INNER JOIN frpm ON satscores.cds = frpm.cdscode WHERE satscores.avgscrread = ( SELECT max(avgscrread) FROM satscores )

**Gold SQL:** SELECT T2.'FRPM Count (Ages 5-17)' FROM satscores AS T1 INNER JOIN frpm AS T2 ON T1.cds = T2.CDSCode ORDER BY T1.AvgScrRead DESC LIMIT 1

Table 27: The evidence information is ignored

**Question:** When did the first-through-twelfth-grade school with the largest enrollment open?
**Evidence:** K-12 means First-through-twelfth-grade;

**Predicted SQL:** SELECT schools.opendate FROM schools INNER JOIN frpm ON schools.cdscode = frpm.cdscode WHERE frpm.'school type' = 'First-through-twelfth-grade' ORDER BY frpm.'enrollment (ages 5-17)' DESC LIMIT 1

**Gold SQL:** SELECT T2.OpenDate FROM frpm AS T1 INNER JOIN schools AS T2 ON T1.CDSCode = T2.CDSCode ORDER BY T1.'Enrollment (K-12)' DESC LIMIT 1

Table 28: Joining additional tables

**Question:** In Los Angeles how many schools have more than 500 free meals but less than 700 free or reduced price meals for K-12?
**Evidence:** None

**Predicted SQL:** SELECT count(schools.school) FROM schools INNER JOIN frpm ON schools.cdscode = frpm.cdscode WHERE frpm.'free meal count (ages 5-17)' > 500 AND frpm.'free meal count (ages 5-17)' < 700 AND schools.city = 'Los Angeles'

**Gold SQL:** SELECT COUNT(CDSCode) FROM frpm WHERE 'County Name' = 'Los Angeles' AND 'Free Meal Count (K-12)' > 500 AND 'Free Meal Count (K-12)' < 700

Table 29: Added additional operations

**Question:** List the names of schools with more than 30 difference in enrollements between K-12 and ages 5-17? Please also give the full street adress of the schools.
**Evidence:** Diffrence in enrollement = 'enrollment (k-12)' - 'enrollment (ages 5-17)';

**Predicted SQL:** SELECT schools.school, schools.StreetAbr FROM schools INNER JOIN frpm ON schools.cdscode = frpm.cdscode WHERE abs(frpm.'enrollment (ages 5-17)' - frpm.'enrollment (k-12)') > 30

**Gold SQL:** SELECT T1.School, T1.StreetAbr FROM schools AS T1 INNER JOIN frpm AS T2 ON T1.CDSCode = T2.CDSCode WHERE T2.'Enrollment (K-12)' - T2.'Enrollment (Ages 5-17)' > 30

Table 30: Incorrect format

**Question:** What is the average score in writing for the schools that were opened after 1991 or closed before 2000? List the school names along with the score. Also, list the communication number of the schools if there is any.
**Evidence:** Communication number refers to phone number.;

**Predicted SQL:** SELECT avg(satscores.avgscrwrite), schools.school, schools.phone FROM schools INNER JOIN satscores ON schools.cdscode = satscores.cds WHERE schools.opendate > '1991-12-31' AND schools.closeddate < '2000-12-31'

**Gold SQL:** SELECT T2.School, T1.AvgScrWrite, T2.Phone, strftime('%Y', T2.OpenDate), strftime('%Y', T2.ClosedDate) FROM schools AS T2 LEFT JOIN satscores AS T1 ON T2.CDSCode = T1.cds WHERE strftime('%Y', T2.OpenDate) > '1991' AND strftime('%Y', T2.ClosedDate) < '2000'

Table 31: Value Error

**Question:** Please list the zip code of all the charter schools in Fresno County Office of Education.
**Evidence:** charter schools refers to 'charter school (y/n)' = 1 in the table fprm;

**Predicted SQL:** SELECT schools.zip FROM schools INNER JOIN frpm ON schools.cdscode = frpm.cdscode WHERE schools.county = 'Fresno' AND frpm.'charter school (y/n)' = 1

**Gold SQL:** SELECT T2.Zip FROM frpm AS T1 INNER JOIN schools AS T2 ON T1.CDSCode = T2.CDSCode WHERE T1.'District Name' = 'Fresno County Office of Education' AND T1.'Charter School (Y/N)' = 1

Table 32: Ignore the underlying information in the question

**Question:** State the names and full communication address of high schools in Monterey which has more than 800 free or reduced price meals for ages 15-17?
**Evidence:** Full communication address should include zip, street, city, state;

**Predicted SQL:** SELECT schools.school, schools.zip, schools.street, schools.city, schools.state FROM schools INNER JOIN frpm ON schools.cdscode = frpm.cdscode WHERE frpm.'free meal count (ages 15-17)' > 800 AND schools.city = 'Monterey'

**Gold SQL:** SELECT T1.'School Name', T2.Zip, T2.Street, T2.City, T2.State FROM frpm AS T1 INNER JOIN schools AS T2 ON T1.CDSCode = T2.CDSCode WHERE T2.County = 'Monterey' AND T1.'Free Meal Count (Ages 5-17)' > 800 AND T1.'School Type' = 'High Schools (Public)'

Table 33: Creating new incorrect constraints. Corrector introduces a new WHERE constraint, but this is wrong.

---

**Question:** What is the percentage of the customers who used EUR in 2012/8/25?
**Evidence:** '2012/8/25' can be represented by '2012-08-25';

---

**Generator's beam search:**
SELECT cast(sum(CASE WHEN customers.currency = 'EUR' THEN 1 ELSE 0 END) AS REAL) * 100 / count(customers.customerid) FROM customers INNER JOIN transactions_1k ON customers.customerid = transactions_1k.customerid WHERE transactions_1k.date = '2012-08-25'
SELECT cast(sum(CASE WHEN customers.currency = 'EUR' THEN 1 ELSE 0 END) AS REAL) * 100 / count(customers.customerid) FROM customers INNER JOIN transactions_1k ON customers.customerid = transactions_1k.customerid WHERE transactions_1k.'date' = '2012-08-25'
SELECT cast(count(CASE WHEN customers.currency = 'EUR' THEN 1 ELSE NULL END) AS REAL) * 100 / count(customers.customerid) FROM customers INNER JOIN transactions_1k ON customers.customerid = transactions_1k.customerid WHERE transactions_1k.date = '2012-08-25'
SELECT cast(count(CASE WHEN customers.currency = 'EUR' THEN 1 ELSE NULL END) AS REAL) * 100 / count(customers.customerid) FROM customers INNER JOIN transactions_1k ON customers.customerid = transactions_1k.customerid WHERE transactions_1k.'date' = '2012-08-25'

---

**Low confidence constraints:**
Where : ( transactions_1k.date = '2012-08-25' | transactions_1k.'date' = '2012-08-25' );
Select : ( CAST(SUM(CASE WHEN customers.currency = 'EUR' THEN 1 ELSE 0 END) AS FLOAT) * 100 | CAST(COUNT(CASE WHEN customers.currency = 'EUR' THEN 1 ELSE NULL END) AS FLOAT) * 100 );

---

**Basic SQL:**
SELECT COUNT(customers.customerid) FROM customers INNER JOIN transactions_1k ON customers.customerid = transactions_1k.customerid

---

**Corrector result SQL:**
SELECT cast(sum(iif(transactions_1k.'date' = '2012-08-25', 1, 0)) AS REAL) * 100 / count(transactions_1k.customerid) FROM transactions_1k INNER JOIN customers ON transactions_1k.customerid = customers.customerid WHERE customers.currency = 'EUR'

---

**Gold SQL:**
SELECT CAST(SUM(IIF(T2.Currency = 'EUR', 1, 0)) AS FLOAT) * 100 / COUNT(T1.CustomerID) FROM transactions_1k AS T1 INNER JOIN customers AS T2 ON T1.CustomerID = T2.CustomerID WHERE T1.Date = '2012-08-25'

---

Table 34: Loss of Original Constraints. Corrector loses the "SELECT 'District Code'" constraint in the correct SQL.

| |
|---|
| **Question:** What is the Percent (%) Eligible Free (K-12) in the school administered by an administrator whose first name is Alusine. List the district code of the school. |
| **Evidence:** percent (%) eligible free (k-12) = 'free meal count (k-12)' / 'enrollment (k-12)' * 100%; |

**Generator's beam search:**
SELECT cast(frpm.'free meal count (k-12)' AS REAL) * 100 / frpm.'enrollment (k-12)', frpm.'district code' FROM schools INNER JOIN frpm ON schools.cdscode = frpm.cdscode WHERE schools.admfname1 = 'Alusine'
SELECT cast(frpm.'free meal count (k-12)' AS REAL) * 100 / frpm.'enrollment (k-12)', frpm.'district code' FROM frpm INNER JOIN schools ON frpm.cdscode = schools.cdscode WHERE schools.admfname1 = 'Alusine'
SELECT cast(frpm.'free meal count (k-12)' AS REAL) * 100 / frpm.'enrollment (k-12)', frpm.'district code' FROM schools INNER JOIN frpm ON schools.cdscode = frpm.cdscode WHERE schools.admfname1 = 'Alusine'
SELECT cast(frpm.free meal count (k-12) AS REAL) * 100 / frpm.'enrollment (k-12)', frpm.'district code' FROM frpm INNER JOIN schools ON frpm.cdscode = schools.cdscode WHERE schools.admfname1 = 'Alusine'

**Low confidence constraints:**
**Select:** ( CAST(frpm.'free meal count (k-12)' AS FLOAT) * 100 / frpm.'enrollment (k-12)' | CAST(frpm.'free meal count (k-12)' AS FLOAT) * 100 / frpm.'enrollment (k-12)' | frpm.'District Code' );

**Basic SQL:**
SELECT * FROM schools INNER JOIN frpm ON schools.cdscode = frpm.cdscode WHERE schools.admfname1 = 'Alusine'

**Corrector result SQL:**
SELECT cast(frpm.'free meal count (k-12)' AS REAL) * 100 / frpm.'enrollment (k-12)' FROM schools INNER JOIN frpm ON schools.cdscode = frpm.cdscode WHERE schools.admfname1 = 'Alusine'

**Gold SQL:**
SELECT T1.'Free Meal Count (K-12)' * 100 / T1.'Enrollment (K-12)', T1.'District Code' FROM frpm AS T1 INNER JOIN schools AS T2 ON T1.CDSCode = T2.CDSCode WHERE T2.AdmFName1 = 'Alusine'

Table 35: Non-conformity to Basic SQL. Corrector does not conform to Basic SQL and loses the LIMIT constraint.

| |
|---|
| **Question:** What is the administrator's email address of the chartered school with the fewest students enrolled in grades 1 through 12?"
**Evidence:** Charted school means 'charter school (y/n)' = 1 in the table frpm; Students enrolled in grades 1 through 12 refers to 'enrollment (k-12)'; |
| **Generator's beam search:**
SELECT schools.admemail1 FROM schools INNER JOIN frpm ON schools.cdscode = frpm.cdscode WHERE frpm.'charter school (y/n)' = 1 ORDER BY frpm.'enrollment (k-12)' LIMIT 1
SELECT schools.admemail1 FROM schools INNER JOIN frpm ON schools.cdscode = frpm.cdscode WHERE frpm.'charter school (y/n)' = 1 ORDER BY frpm.'enrollment (k-12)' ASC LIMIT 1
SELECT schools.admemail1 FROM frpm INNER JOIN schools ON frpm.cdscode = schools.cdscode WHERE schools.'charter school (y/n)' = 1 ORDER BY frpm.'enrollment (k-12)' LIMIT 1
SELECT schools.admemail1 FROM schools INNER JOIN frpm ON schools.cdscode = frpm.cdscode WHERE frpm.'charter school (y/n)' = 1 ORDER BY frpm.'enrollment (ages 5-17)' LIMIT 1 |
| **Low confidence constraints:**
**Order:** ( ORDER BY frpm.'enrollment (k-12)' | ORDER BY frpm.'enrollment (k-12)' ASC | ORDER BY frpm.'enrollment (ages 5-17)' );
**Where:** ( frpm.'charter school (y/n)' = 1 | schools.'charter school (y/n)' = 1 ); |
| **Basic SQL:**
SELECT schools.admemail1 FROM schools INNER JOIN frpm ON schools.cdscode = frpm.cdscode LIMIT 1 |
| **Corrector result SQL:**
ELECT schools.admemail1 FROM schools INNER JOIN frpm ON schools.cdscode = frpm.cdscode WHERE frpm.'enrollment (k-12)' = ( SELECT min('enrollment (k-12)') FROM frpm ) |
| **Gold SQL:**
SELECT T2.AdmEmail1 FROM frpm AS T1 INNER JOIN schools AS T2 ON T1.CDSCode = T2.CDSCode WHERE T1.'Charter School (Y/N)' = 1 ORDER BY T1.'Enrollment (K-12)' ASC LIMIT 1 |

Table 36: Choosing the Wrong Constraint. Corrector selected the wrong SELECT constraint in low confidence constraints.

| |
|---|
| **Question:** For the driver who set the fastest lap speed in race No.933, where does he come from? **Evidence:** fastest lap speed refers to MAX(fastestlapspeed); |
| **Generator's beam search:** SELECT drivers.nationality FROM drivers INNER JOIN results ON drivers.driverid = results.driverid WHERE results.raceid = 933 ORDER BY results.fastestlapspeed DESC LIMIT 1 SELECT drivers.forename, drivers.surname FROM drivers INNER JOIN results ON drivers.driverid = results.driverid WHERE results.raceid = 933 ORDER BY results.fastestlapspeed DESC LIMIT 1 SELECT drivers.forename, drivers.surname FROM results INNER JOIN drivers ON results.driverid = drivers.driverid WHERE results.raceid = 933 ORDER BY results.fastestlapspeed DESC LIMIT 1 SELECT drivers.nationality FROM results INNER JOIN drivers ON results.driverid = drivers.driverid WHERE results.raceid = 933 ORDER BY results.fastestlapspeed DESC LIMIT1 |
| **Low confidence constraints:** **Select:** ( drivers.nationality \| ('drivers.forename', 'drivers.surname') ); |
| **Basic SQL:** SELECT * FROM drivers INNER JOIN results ON drivers.driverid = results.driverid WHERE results.raceid = 933 ORDER BY results.fastestlapspeed DESC LIMIT 1 |
| **Corrector result SQL:** SELECT drivers.forename, drivers.surname FROM drivers INNER JOIN results ON drivers.driverid = results.driverid WHERE results.raceid = 933 ORDER BY results.fastestlapspeed DESC LIMIT 1 |
| **Gold SQL:** SELECT T1.nationality FROM drivers AS T1 INNER JOIN results AS T2 ON T2.driverId = T1.driverId WHERE T2.raceId = 933 AND T2.fastestLapTime IS NOT NULL ORDER BY T2.fastestLapSpeed DESC LIMIT 1 |

Table 37: Adding Additional Error Constraints. The original result of the Generator had no extra constraints, but the Corrector added.

| |
|---|
| **Question:** How many members of the Student_Club have majored Environmental Engineering? 
 **Evidence:** 'Environmental Engineering' is the major name; |
| **Generator's beam search:** 
 SELECT count(member.member_id) FROM member INNER JOIN major ON member.link_to_major = major.major_id WHERE major.major_name = 'Environmental Engineering' 
 SELECT count(DISTINCT member.member_id) FROM member INNER JOIN major ON member.link_to_major = major.major_id WHERE major.major_name = 'Environmental Engineering' 
 SELECT count(member.member_id) FROM member INNER JOIN major ON member.link_to_major = major.major_id WHERE major.major_name = 'Environmental Engineering' AND member.position = 'Student_Club' 
 SELECT count(member.first_name) FROM member INNER JOIN major ON member.link_to_major = major.major_id WHERE major.major_name = 'Environmental Engineering' |
| **Low confidence constraints:** 
 **Select :** ( COUNT(member.member_id) | COUNT(DISTINCT member.member_id) | COUNT(member.first_name) ); **Where :** ( member.position = 'Student_Club' ); |
| **Basic SQL:** 
 SELECT * FROM member INNER JOIN major ON member.link_to_major = major.major_id WHERE major.major_name = 'Environmental Engineering' |
| **Corrector result SQL:** 
 SELECT count(member.member_id) FROM member INNER JOIN major ON member.link_to_major = major.major_id WHERE major.major_name = 'Environmental Engineering' AND member.position = 'Student_Club' |
| **Gold SQL:** 
 SELECT COUNT(T1.member_id) FROM member AS T1 INNER JOIN major AS T2 ON T1.link_to_major = T2.major_id WHERE T2.major_name = 'Environmental Engineering' |

Table 38: Choosing the Right Constraints. Corrector selects the correct constraint among the low confidence constraints

| |
|---|
| **Question:** What is the unabbreviated mailing address of the school with the highest FRPM count for K-12 students? 
 **Evidence:** None; |
| **Generator's beam search:** 
 SELECT schools.mailstrabr FROM schools INNER JOIN frpm ON schools.cdscode = frpm.cdscode ORDER BY frpm.'free meal count (k-12)' DESC LIMIT 1 
 SELECT schools.mailstreet FROM schools INNER JOIN frpm ON schools.cdscode = frpm.cdscode ORDER BY frpm.'free meal count (k-12)' DESC LIMIT 1 
 SELECT schools.mailstrabr FROM schools INNER JOIN frpm ON schools.cdscode = frpm.cdscode ORDER BY frpm.'frpm count (k-12)' DESC LIMIT 1 
 SELECT schools.mailstrabr FROM schools INNER JOIN frpm ON schools.cdscode = frpm.cdscode ORDER BY frpm.'free meal count (k-12)' DESC LIMIT 1 |
| **Low confidence constraints:** 
 Select : ( schools.mailstrabr | schools.mailstreet ); 
 Order : ( ORDER BY frpm.'free meal count (k-12)' DESC | ORDER BY frpm.'frpm count (k-12)' DESC | ORDER BY frpm.'free meal count (k-12)' DESC ); |
| **Basic SQL:** 
 SELECT * FROM schools INNER JOIN frpm ON schools.cdscode = frpm.cdscode LIMIT 1 |
| **Corrector result SQL:** 
 SELECT DISTINCT schools.mailstreet FROM schools INNER JOIN frpm ON schools.cdscode = frpm.cdscode ORDER BY frpm.'frpm count (k-12)' DESC LIMIT 1 |
| **Generator's wrong SQL:** 
 SELECT schools.mailstrabr FROM schools INNER JOIN frpm ON schools.cdscode = frpm.cdscode ORDER BY frpm.'free meal count (k-12)' DESC LIMIT 1 |

Table 39: Creating new correct constraints. Corrector adds a SELECT constraint that was not present before.

| |
|---|
| **Question:** What is the average writing score of each of the schools managed by Ricci Ulrich? List the schools and the corresponding average writing scores. 
 **Evidence:** Usually, administrators manage the school stuff.; |
| **Generator's beam search:** 
 SELECT avg(satscores.avgscrwrite) FROM schools INNER JOIN satscores ON schools.cdscode = satscores.cds WHERE schools.admfname1 = 'Ricci' AND schools.admlname1 = 'Ulrich' 
 SELECT avg(satscores.avgscrwrite) FROM schools INNER JOIN satscores ON schools.cdscode = satscores.cds WHERE schools.admfname1 = 'Ricci' AND schools.admlname1 = 'Ulrich' 
 SELECT avg(satscores.avgscrwrite) FROM satscores INNER JOIN schools ON satscores.cds = schools.cdscode WHERE schools.admfname1 = 'Ricci' AND schools.admlname1 = 'Ulrich' 
 SELECT cast(avg(satscores.avgscrwrite) AS REAL) FROM schools INNER JOIN satscores ON schools.cdscode = satscores.cds WHERE schools.admfname1 = 'Ricci' AND schools.admlname1 = 'Ulrich' |
| **Low confidence constraints:** 
 **Select :** ( AVG(satscores.avgscrwrite) \| CAST(AVG(satscores.avgscrwrite) AS FLOAT) ); |
| **Basic SQL:** 
 SELECT * FROM schools INNER JOIN satscores ON schools.cdscode = satscores.cds WHERE schools.admfname1 = 'Ricci' AND schools.admlname1 = 'Ulrich' |
| **Corrector result SQL:** 
 SELECT schools.school, avg(satscores.avgscrwrite) FROM schools INNER JOIN satscores ON schools.cdscode = satscores.cds WHERE schools.admfname1 = 'Ricci' AND schools.admlname1 = 'Ulrich' GROUP BY schools.school |
| **Generator's wrong SQL:** 
 SELECT avg(satscores.avgscrwrite) FROM schools INNER JOIN satscores ON schools.cdscode = satscores.cds WHERE schools.admfname1 = 'Ricci' AND schools.admlname1 = 'Ulrich' |

Table 40: Deleting Incorrect Constraints. The Corrector removes the WHERE constraint from the results in the Generator.

---

**Question:** Please list the phone numbers of the schools with the top 3 SAT excellence rate.
**Evidence:** Excellence rate = numge1500 / numtsttakr;

---

**Generator's beam search:**
SELECT schools.phone FROM schools INNER JOIN satscores ON schools.cdscode = satscores.cds WHERE cast(satscores.numge1500 AS REAL) * 100 / satscores.numtsttakr = ( SE-LECT max(cast(numge1500 AS REAL) * 100 / numtsttakr) FROM satscores ) LIMIT 3
SELECT schools.phone FROM schools INNER JOIN satscores ON schools.cdscode = satscores.cds WHERE cast(satscores.numge1500 AS REAL) / satscores.numtsttakr = ( SELECT max(cast(numge1500 AS REAL) / numtsttakr) FROM satscores ) LIMIT 3
SELECT schools.phone FROM satscores INNER JOIN schools ON satscores.cds = schools.cdscode ORDER BY cast(satscores.numge1500 AS REAL) / satscores.numtsttakr DESC LIMIT 3
SELECT schools.phone FROM satscores INNER JOIN schools ON satscores.cds = schools.cdscode ORDER BY satscores.numge1500 / satscores.numtsttakr DESC LIMIT 3

---

**Low confidence constraints:**
Where : ( CAST(satscores.numge1500 AS FLOAT) * 100 / satscores.numtsttakr | (SELECT MAX(CAST(numge1500 AS FLOAT) * 100 / numtsttakr) FROM satscores) | CAST(satscores.numge1500 AS FLOAT) / satscores.numtsttakr | (SELECT MAX(CAST(numge1500 AS FLOAT) / numtsttakr) FROM satscores) );
Order : ( ORDER BY CAST(satscores.numge1500 AS FLOAT) / satscores.numtsttakr DESC | ORDER BY satscores.numge1500 / satscores.numtsttakr DESC );

---

**Basic SQL:**
SELECT schools.phone FROM schools INNER JOIN satscores ON schools.cdscode = satscores.cds LIMIT 3

---

**Corrector result SQL:**
SELECT schools.phone FROM schools INNER JOIN satscores ON schools.cdscode = satscores.cds ORDER BY cast(satscores.numge1500 AS REAL) / satscores.numtsttakr DESC LIMIT 3

---

**Generator's wrong SQL:**
SELECT schools.phone FROM schools INNER JOIN satscores ON schools.cdscode = satscores.cds WHERE cast(satscores.numge1500 AS REAL) * 100 / satscores.numtsttakr = ( SELECT max(cast(numge1500 AS REAL) * 100 / numtsttakr) FROM satscores ) LIMIT 3

---

Table 41: Modifying Potential Error Constraints in Basic SQL. Additional tables were connected in basic SQL, and Corrector corrected it.

| |
|---|
| **Question:** List all patients with their sex and date of birthday, whose AST glutamic oxaloacetic transaminase (GOT) index is within normal range for loboratory examination in 1994. **Evidencd:** AST glutamic oxaloacetic transaminase (got) index is within normal range refers to got < 60; examination in 1994 refers to year(date) = 1994; |
| **Generator's beam search:** ELECT patient.sex, patient.birthday FROM laboratory INNER JOIN patient ON laboratory.id = patient.id INNER JOIN examination ON patient.id = examination.id WHERE strftime('%Y', laboratory.'date') = '1994' AND laboratory.got < 60 SELECT patient.sex, patient.birthday FROM laboratory INNER JOIN patient ON laboratory.id = patient.id INNER JOIN examination ON patient.id = examination.id WHERE strftime('%Y', laboratory.date) = '1994' AND laboratory.got < 60 SELECT DISTINCT patient.sex, patient.birthday FROM laboratory INNER JOIN patient ON laboratory.id = patient.id INNER JOIN examination ON patient.id = examination.id WHERE strftime('%Y', laboratory.'date') = '1994' AND laboratory.got < 60 SELECT DISTINCT patient.sex, patient.birthday FROM laboratory INNER JOIN patient ON laboratory.id = patient.id INNER JOIN examination ON patient.id = examination.id WHERE strftime('%Y', laboratory.date) = '1994' AND laboratory.got < 60 |
| **Where :** ( STRFTIME('%Y', laboratory.'date') = '1994' \| STRFTIME('%Y', laboratory.date) = '1994' ); Select : ( DISTINCT ); |
| **Basic SQL:** SELECT patient.sex, patient.birthday FROM laboratory INNER JOIN patient ON laboratory.id = patient.id INNER JOIN examination ON patient.id = examination.id WHERE laboratory.got < 60 |
| **Corrector result SQL:** SELECT patient.sex, patient.birthday FROM patient INNER JOIN laboratory ON patient.id = laboratory.id WHERE strftime('%Y', laboratory.'date') = '1994' AND laboratory.got < 60 |
| **Generator's wrong SQL:** ELECT patient.sex, patient.birthday FROM laboratory INNER JOIN patient ON laboratory.id = patient.id INNER JOIN examination ON patient.id = examination.id WHERE strftime('%Y', laboratory.'date') = '1994' AND laboratory.got < 60 |

**Input SQL**

"SELECT DISTINCT T1.AdmFName1, T1.District FROM schools AS T1 INNER JOIN ( SELECT admfname1 FROM schools GROUP BY admfname1 ORDER BY COUNT(admfname1) DESC LIMIT 2 ) AS T2 ON T1.AdmFName1 = T2.admfname1"

- - - - - - - - - - - - - - - - - - - - - - - - - - - - - - - - - - - - - - - - - - - - - -

**Output**

**SQL Sub-queries**
**Sub-query 1**: "SELECT admfname1 FROM schools GROUP BY admfname1 ORDER BY COUNT (admfname1) DESC LIMIT 2",
**Sub-query 2**: "SELECT DISTINCT T1.AdmFName1, T1.District FROM schools AS T1 INNER JOIN (SELECT admfname1 FROM schools GROUP BY admfname1 ORDER BY COUNT (admfname1) DESC LIMIT 2) AS T2 ON T1.AdmFName1 = T2.admfname1"

- - - - - - - - - - - - - - - - - - - - - - - - - - - - - - - - - - - - - - - - - - - - - -

**Decomposed sub-SQLs**
**Sub-query 1**: {
"1": "SELECT * FROM schools",
"2": "SELECT * FROM schools GROUP BY admfname1",
"3": "SELECT * FROM schools LIMIT 2",
"4": "SELECT admfname1 FROM schools",
"5": "SELECT * FROM schools GROUP BY admfname1 ORDER BY COUNT (admfname1) DESC",
"6": "SELECT * FROM schools GROUP BY admfname1 LIMIT 2",
"7": "SELECT admfname1 FROM schools GROUP BY admfname1",
"8": "SELECT admfname1 FROM schools LIMIT 2",
"9": "SELECT * FROM schools GROUP BY admfname1 ORDER BY COUNT (admfname1) DESC LIMIT 2",
"10": "SELECT admfname1 FROM schools GROUP BY admfname1 ORDER BY COUNT (admfname1) DESC",
"11": "SELECT admfname1 FROM schools GROUP BY admfname1 LIMIT 2",
"12": "SELECT admfname1 FROM schools GROUP BY admfname1 ORDER BY COUNT (admfname1) DESC LIMIT 2" },
**Sub-query 2**: {
"1": "SELECT * FROM schools AS T1",
"2": "SELECT * FROM schools AS T1 INNER JOIN (SELECT admfname1 FROM schools GROUP BY admfname1 ORDER BY COUNT (admfname1) DESC LIMIT 2) AS T2 ON T1.AdmFName1 = T2.admfname1",
"3": "SELECT DISTINCT * FROM schools AS T1",
"4": "SELECT DISTINCT * FROM schools AS T1 INNER JOIN (SELECT admfname1 FROM schools GROUP BY admfname1 ORDER BY COUNT (admfname1) DESC LIMIT 2) AS T2 ON T1.AdmFName1 = T2.admfname1",
"5": "SELECT DISTINCT T1.AdmFName1, T1.District FROM schools AS T1",
"6": "SELECT DISTINCT T1.AdmFName1, T1.District FROM schools AS T1 INNER JOIN (SELECT admfname1 FROM schools GROUP BY admfname1 ORDER BY COUNT(admfname1) DESC LIMIT 2) AS T2 ON T1.AdmFName1 = T2.admfname1" }

- - - - - - - - - - - - - - - - - - - - - - - - - - - - - - - - - - - - - - - - - - - - - -

**Reasoning Paths**
**Sub-query 1**: {
"p1": [1, 2, 5, 9, 12],      "p2": [1, 2, 5, 10, 12],      "p3": [1, 2, 6, 9, 12],
"p4": [1, 2, 6, 11, 12],      "p5": [1, 2, 7, 10, 12],      "p6": [1, 2, 7, 11, 12],
"p7": [1, 3, 6, 9, 12],      "p8": [1, 3, 6, 11, 12],      "p9": [1, 3, 8, 11, 12],
"p10": [1, 4, 7, 10, 12],      "p11": [1, 4, 7, 11, 12],      "p12": [1, 4, 8, 11, 12] },

**Sub-query 2**: {
"p1": [1, 2, 4, 6],      "p2": [1, 3, 4, 6],      "p3": [1, 3, 5, 6] }

Figure 8: An example of input and output in READER

**Instructions**
You are an expert at translating SQL queries into natural language questions. Your task is to generate a clear, concise, and detailed question that accurately captures the intent of the SQL query.

- - - - - - - - - - - - - - - - - - - - - - - - - - - - - - - - - - - - - - - - - - - - - -

**Example**
**SQL Query:** SELECT avg(ratings.rating_score) FROM movies INNER JOIN ratings ON movies.movie_id = ratings.movie_id WHERE movies.movie_title = 'When Will I Be Loved'
**Generated Question:** What is the average rating for movie titled 'When Will I Be Loved'?

**SQL Query:** SELECT products.name FROM products INNER JOIN sales ON products.productid = sales.productid WHERE sales.salespersonid = 20 ORDER BY sales.quantity DESC LIMIT 1
**Generated Question:** What is the name of the product that is most sold by sale person id 20?

**SQL Query:** {Origin SQL}
**Generated Question:** {Origin question}

- - - - - - - - - - - - - - - - - - - - - - - - - - - - - - - - - - - - - - - - - - - - - -

Now, create a detailed yet concise question that is semantically consistent with the following SQL query. Ensure that the generated question closely follows the structure of the example, while accounting for any differences in the SQL.

**SQL Query:** {Sub-SQL}
**Generated Question:**

Figure 9: Sub-question generation prompt for the augmented (sub-question, sub-SQL) pairs

**Original (Question,SQL) pair**
Question: What are the URL to the list page on Mubi of the lists with followers between 1-2 and whose last update timestamp was on 2012?
SQL: SELECT list_url FROM LISTS WHERE list_update_timestamp_utc LIKE '2012%' AND list_followers BETWEEN 1 AND 2 ORDER BY list_update_timestamp_utc DESC LIMIT 1

- - - - - - - - - - - - - - - - - - - - - - - - - - - - - - - - - - - - - - - - - - - - - -

**(sub-question, sub-SQL) pairs**

**Pair #1:**
**sub-question:** What are the URLs of the lists on Mubi with a last update timestamp in 2012 and a follower count between 1 and 2, sorted by the update timestamp in descending order?
**sub-SQL:** SELECT list_url FROM LISTS WHERE list_update_timestamp_utc LIKE '2012%' AND list_followers BETWEEN 1 AND 2 ORDER BY list_update_timestamp_utc DES

**Pair #2:**
**sub-question:** What is the URL of the list page on Mubi with the fewest followers, where the last update timestamp is the most recent?
**sub-SQL:** SELECT list_url FROM LISTS WHERE list_followers BETWEEN 1 AND 2 ORDER BY list_update_timestamp_utc DESC LIMIT 1

Figure 10: An example of a (sub-question, sub-SQL) pair

**Instructions**
Generate a chain-of-thought (CoT) reasoning process that explains how each sub-SQL query incrementally builds towards answering the original question. Ensure the reasoning is clear, concise, and logically follows the progression of the sub-SQLs.

- - - - - - - - - - - - - - - - - - - - - - - - - - - - - - - - - - - - - - - - - - - -

**Examples**
**Question:** What is the average writing score of each of the schools managed by Ricci Ulrich? List the schools and the corresponding average writing scores.
**Sub-SQLs list:** /**
1.SELECT * FROM satscores
2.SELECT * FROM satscores INNER JOIN schools ON satscores.cds = schools.cdscode
3.SELECT * FROM satscores INNER JOIN schools ON satscores.cds = schools.cdscode HWERE schools.admlname1 = 'Ulrich'
4.SELECT * FROM satscores INNER JOIN schools ON satscores.cds = schools.cdscode WHERE schools.admfname1 = 'Ricci' AND schools.admlname1 = 'Ulrich'
5.SELECT satscores.avgscrwrite FROM satscores INNER JOIN schools ON satscores.cds = schools.cdscode WHERE schools.admfname1 = 'Ricci' AND schools.admlname1 = 'Ulrich' **/

Now, the requirements are as follows: the output must be short and consist of a sentence, each sub-SQL generates half of a sentence, split by ',', only the reasoning process needs to be output and output in English:

**Generate reasoning path:**
First, select information from satscores, then join the schools table, add school information, and add a filter based on Ricci Ulrich; finally, choose to display the average writing score.

- - - - - - - - - - - - - - - - - - - - - - - - - - - - - - - - - - - - - - - - - - - -

I will provide a new question and sub-SQLs list. Following the above example, generate the corresponding reasoning path step-by-step.

**Question:** {Question}
**Sub-SQLs list:** /**
{Sub-SQLs list} **/

Now, The requirements are as follows: the output must be short and consist of a sentence, each sub-SQL generates half of a sentence, split by ',', only the reasoning process needs to be output and output in English:

**Generate reasoning path:**

Figure 11: Reason generation prompt

**Reasoning path**
Sub-SQL 1 : SELECT * FROM LISTS,
Sub-SQL 2 : SELECT * FROM LISTS ORDER BY list_update_timestamp_utc DESC,
Sub-SQL 3 : SELECT * FROM LISTS WHERE list_followers BETWEEN 1 AND 2 OR-
DER BY list_update_timestamp_utc DESC,
Sub-SQL 4 : SELECT * FROM LISTS WHERE list_update_timestamp_utc LIKE '2012%'
AND list_followers BETWEEN 1 AND 2 ORDER BY list_update_timestamp_utc DESC,
Sub-SQL 5 : SELECT * FROM LISTS WHERE list_update_timestamp_utc LIKE '2012%'
AND list_followers BETWEEN 1 AND 2 ORDER BY list_update_timestamp_utc DESC
LIMIT 1,
Sub-SQL 6 : SELECT list_url FROM LISTS WHERE list_update_timestamp_utc LIKE
'2012%' AND list_followers BETWEEN 1 AND 2 ORDER BY list_update_timestamp_utc
DESC LIMIT 1

- - - - - - - - - - - - - - - - - - - - - - - - - - - - - - - - - - - - - - - - - - - -

**(question, reason) pair**

**question:** What are the URL to the list page on Mubi of the lists with followers between 1-2
and whose last update timestamp was on 2012?
**reason:** Select all lists, then sort by update timestamp in descending order; filter for lists
with followers between 1-2 and an update timestamp in 2012; Keep the order and limit it to
the first item; retrieve the URL for the specified list.

Figure 12: An example of a (question, reason) pair, where the reason is generated from the given
reasoning path for the question

**SYSTEM_SELF_CORRECTION_PROMPT**

For the given question, use the provided tables, columns, foreign keys, and primary keys to fix the given SQLite SQL QUERY for any issues. If there are any problems, fix them. If there are no issues, return the SQLite SQL QUERY as is. Hint helps you to write the correct sqlite SQL query.

Use the following instructions for fixing the sqlite SQL query:

1) Avoid redundant columns in SELECT clause, all of the columns should be mentioned in the question.
2) Pay attention to the columns that are used for the JOIN by checking the Foreign keys.
3) Pay attention to the columns that are used for the WHERE statement.
4) Pay attention to the columns that are used for the GROUP BY statement.
5) Pay attention to the columns that are used for the ORDER BY statement.
6) check that all of the columns exist in the table and there are no typos.
7) Use CAST when is needed.
8) USE CASE WHEN is needed.

- - - - - - - - - - - - - - - - - - - - - - - - - - - - - - - - - - - - - - - - - - -

**Few examples of this task are:**
Schema of the database with sample rows and column descriptions:
CREATE TABLE movies ( movie_id INTEGER NOT NULL, movie_release_year INTE-GER, ... ...)
**3 rows from movies table:**
movie_id movie_title movie_release_year movie_url ... ...
**Table: movies**
Column movie_id: column description -> ID related to the movie on Mubi
Column movie_title: column description -> Name of the movie
Column movie_release_year: column description -> Release year of the movie
Column movie_url: column description -> URL to the movie page on Mubi
**Q:** Name movie titles released in year 1945. Sort the listing by the descending order of movie popularity.
**Hint:** released in the year 1945 refers to movie_release_year = 1945;
**SQL:** SELECT movie_title, movie_popularity FROM movies WHERE movie_release_year = 1945/01/01 ORDER BY movie_popularity DESC LIMIT 1
**A: Let's think step by step to find the correct answer.**
1) The column movie_popularity is not mentioned in the question so it's redundant.
2) JOIN is not required as there is no need to join any tables.
3) The condition movie_release_year = 1945/01/01 is not correct. The correct condition is movie_release_year = 1945.
4) GROUP BY is not required as there is no need to group any columns.
5) The ORDER BY clause is correct.
6) all columns are correct and there are no typo errors.
7) CAST is not required as there is no need to cast any columns. 8) CASE is not required as there is no need to use CASE.
So, the final sqlite SQL query answer to the question the given question is = Revised_SQL: SELECT movie_title FROM movies WHERE movie_release_year = 1945 ORDER BY movie_popularity DESC LIMIT 1

- - - - - - - - - - - - - - - - - - - - - - - - - - - - - - - - - - - - - - - - - - -

Evaluate the correctness of this query for the given question. Hint helps you to write the correct SQL query. Correct it if there are any issues. If there are no issues, return the SQLite SQL QUERY as is. Schema of the database with sample rows and column descriptions:
{schema}
{columns_descriptions}
Q: {question}
Hint: {hint}
SQL: {sql_query}
A: Let's think step by step to find the correct answer.

Figure 13: The DIN-SQL's Self-Correction prompt by GPT-4o, use '...' to replace part of the content to decompress space.

**Instructions**
You are an expert in generating SQL. Your task is to compare the predicted SQL results in beam search and regenerate the correct SQL. I will provide you with the relevant database schema, additional table schema, value matching, beam search results, error execute messages, the basic SQL composed of the same parts, and the evidence related to SQL. Regenerate the SQL corresponding to the question based on this information.

**Field Explanation:**
- Table schema: table_name columns = [ talbe_name.column_name ( column type | column name | examples of value ) ];
- Additional table schema: table_name ( table_name.column_name | column type | column description | value description );
- Value matching: The values that need to be matched in the fuzzy constraints. table_name.column_name ( similar value );
- Beam search results: The results of the beam search.
- Error excute messages: The error message generated by the predicted SQL when executed.
- Basic SQL: In the beam search results, the SQL consists of the same clauses.
- Evidence: The evidence related to the SQL statement.
- Question: The question that needs to be answered by the SQL statement.

- - - - - - - - - - - - - - - - - - - - - - - - - - - - - - - - - - - - - - - - - - - -

**Here is a example**
**Table schema:** table location , columns = [ location.state ( text | values : Elbasan , Tirane ) , location.statecode ( text | values : AL , DZ ) , location.city ( text | values : Elbasan , Tirana ) , location.country ( text | values : Albania , Algeria ) , ... ... ]
**Additional table schema:** table user : ( user.gender | text | user's gender | male / female / unknown ); ... ...
**Value matching:** user.gender ( Male ); location.city ( Hale ); location.city ( Malm ); location.city ( River Vale )
**Beam search results:**
Result 1 : SELECT cast(sum(CASE WHEN gender = 'Male' THEN 1 ELSE 0 END) AS REAL) * 100 / count(*) FROM twitter WHERE sentiment > 0
Result 2 : ... ... Result 3 : ... ... Result 4 : ... ...
**Error execute messages:** no such column: gender;
**Basic SQL:** SELECT * FROM twitter
**Evidence:** positive sentiment refers to sentiment > 0; male user refers to gender = 'Male'; percentage = Divide (Count(tweetid where gender = 'Male'), Count (tweetid)) * 100;
**Question:** Among all the tweets with a positive sentiment, what is the percentage of those posted by a male user?
**Final SQL** : SELECT sum(CASE WHEN USER.gender = 'Male' THEN 1.0 ELSE 0 END) / count(twitter.tweetid) AS per FROM twitter INNER JOIN USER ON twitter.userid = USER.userid WHERE twitter.sentiment > 0

- - - - - - - - - - - - - - - - - - - - - - - - - - - - - - - - - - - - - - - - - - - -

Based on the above example, compare the results of the beam search provided below, You only need to output the final SQL, and don't output the others.
**Table schema:** {Table schema}

**Additional table schema:** {Additional table schema}

**Value matching:** {Value matching}

**Beam search results:** {Different constraints}

**Error execute messages:** {Error execute messages}

**Basic SQL:** {Basic SQL}

**Evidence:** {Evidence}

**Question:** {Question}
**Final SQL** :

Figure 14: The Self-Correction prompt by GPT-4o to replace Corrector in READ-SQL

**Instructions**

You are an expert in generating SQL. Your task is to compare the predicted SQL results in beam search and regenerate the correct SQL. I will provide you with the relevant database schema, Beam search results, and the evidence related to SQL. Regenerate the SQL corresponding to the question based on this information.

**Field Explanation:**

- Table schema: table_name columns = [ talbe_name.column_name ( column type | column name | examples of value ) ];
- Beam search results: The results of the beam search.
- Error excute messages: The error message generated by the predicted SQL when executed.
- Evidence: The evidence related to the SQL statement.
- Question: The question that needs to be answered by the SQL statement.

- - - - - - - - - - - - - - - - - - - - - - - - - - - - - - - - - - - - - - - - - -

**Here is a example**

**Table schema:** table location , columns = [ location.state ( text | values : Elbasan , Tirane ) , location.statecode ( text | values : AL , DZ ) , location.city ( text | values : Elbasan , Tirana ) , location.country ( text | values : Albania , Algeria ) , ... ... ]

**Beam search results:**

Result 1 : SELECT cast(sum(CASE WHEN gender = 'Male' THEN 1 ELSE 0 END) AS REAL) * 100 / count(*) FROM twitter WHERE sentiment > 0

Result 2 : SELECT cast(sum(CASE WHEN user.gender = 'Male' THEN 1 ELSE 0 END) AS REAL) * 100 / count(*) FROM twitter INNER JOIN USER ON twitter.userid = USER.userid WHERE twitter.sentiment > 0

Result 3 : SELECT cast(sum(CASE WHEN user.gender = 'Male' THEN 1 ELSE 0 END) AS REAL) * 100 / count(*) FROM twitter INNER JOIN user ON twitter.userid = user.userid WHERE twitter.sentiment > 0

Result 4 : ... ...

**Error execute messages:** no such column: gender;

**Evidence:** positive sentiment refers to sentiment > 0; male user refers to gender = 'Male'; percentage = Divide (Count(tweetid where gender = 'Male'), Count (tweetid)) * 100;

**Question:** Among all the tweets with a positive sentiment, what is the percentage of those posted by a male user?

**Final SQL** : SELECT sum(CASE WHEN USER.gender = 'Male' THEN 1.0 ELSE 0 END) / count(twitter.tweetid) AS per FROM twitter INNER JOIN USER ON twitter.userid = USER.userid WHERE twitter.sentiment > 0

- - - - - - - - - - - - - - - - - - - - - - - - - - - - - - - - - - - - - - - - - -

Based on the above example, compare the results of the beam search provided below, You only need to output the final SQL, and don't output the others.

**Table schema:** {Table schema}

**Beam search results:** {Different constraints}

**Error execute messages:** {Error execute messages}

**Evidence:** {Evidence}

**Question:** {Question}

**Final SQL** :

Figure 15: The Standard Self-Correction prompt

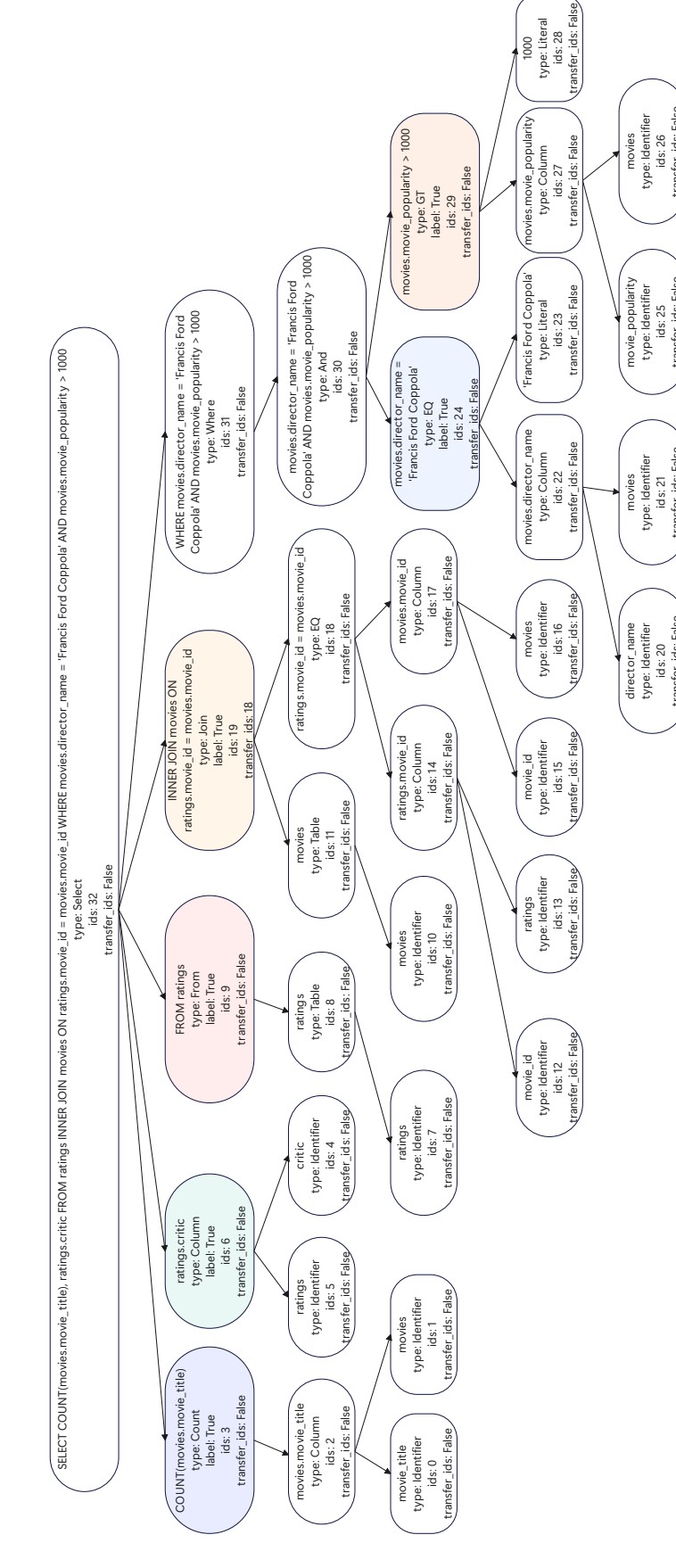

Figure 16: The AST diagram following the READER constraint identification illustrates the attributes of each node: (1) **type** refers to the type of the current node; (2) **label** indicates whether the node is a constraint node. If true, both the node itself and all its child nodes can be considered constraints, which facilitates subsequent deletion; (3) **ids** represents the index of each node; and (4) **transfer_ids** denotes the source of its constraint identifier. The nodes with colored background are constraint nodes.

