# OpenReview forum: "READ-SQL: Reasoning Path Decomposer for Text-to-SQL"
_ICLR.cc/2025/Conference — Submitted to ICLR 2025_

### Official Review · Reviewer_GSF2 · 2024-10-28

**Soundness:** 2
**Presentation:** 2
**Contribution:** 2
**Rating:** 3
**Confidence:** 4

**Summary:**

This paper studied the Text-to-SQL task, and proposed a READ-SQL framework to improve the model's understanding of questions and SQL clauses. READ-SQL parsed the SQL into AST and decomposed into clauses to form the sub-SQLs and reasoning paths. Based on the sub-SQL and reasoning, READ-SQL generated augmented data with the LLM to train a Generator sensitive to constraint changes and reasoning. READ-SQL also trained a Corrector to recognize low-confident clauses for self-correction. Finally, the authors conducted extensive experiments to demonstrate the effectiveness of the proposed method.

**Strengths:**

1. The idea of aligning the natural language and structured syntax to correctly interpret the constraints is interesting.
2. The paper proposed to generate the augmented training data via clause decomposition of SQL, which is novel.
3. The proposed method outperformed leading baselines, and surpassed larger models on BIRD dataset.

**Weaknesses:**

1. The proposed seemed to be an incremental improvement of the existing Text-to-SQL methods, and the technical contributions seemed to be limited.
2. The inference pipeline of the proposed framework is unclear. The paper detailed introduced the data generation process, but it did not show how the trained generator and corrector were used to generate the SQL, i.e., the paper lacked an overall description from the text input to the SQL output.
3. The motivation behind the proposed method should be clarified more clearly. For example, what is the purpose of constraint identification and decomposition, and why can it solve the problems mentioned in the introduction.
4. Some technical introductions are unclear, which makes it difficult to understand the paper.
- What does the constraint or sub-SQL mean, and what can be considered as a constraint?
- What are the goals of the Generator and Corrector, how are they used in SQL generation, and how is the Corrector trained and realized?
- The proposed method regenerated the question based on the sub-SQL, so what is the relation between the generated question and the original ones, and have the authors considered their consistency? Besides, how did the authors ensure that the sub-SQL could solve the generated question?

**Questions:**

See the Weaknesses.

---

> ### Author Response · Authors · 2024-11-25
> **Response to Reviewer GSF2 (Part 1)**
>
> Thank you for your thoughtful feedback and thorough review. Below, we provide responses to your questions and concerns.
>
> > **W1: The proposed seemed to be an incremental improvement of the existing Text-to-SQL methods, and the technical contributions seemed to be limited.**
>
> Thank you for the feedback. We argue that the contribution of READ-SQL lies not only in its exceptional performance but also in its potential to inspire mainstream adoption for real-world Text-to-SQL applications and to generate new ideas for code analysis in other languages. For further explanations, please refer to A1 of the Global Comment.
>
> ---
>
> > **W2: The inference pipeline of the proposed framework is unclear. The paper detailed introduced the data generation process, but it did not show how the trained Generator and corrector were used to generate the SQL, i.e., the paper lacked an overall description from the text input to the SQL output.**
>
> Thank you for your valuable suggestion. We have incorporated your feedback into the revision and updated it in Sec. 3.4 (lines 304-323).  For more details, please refer to A3 of the Global Comment.
>
> ---
>
> > **W3: The motivation behind the proposed method should be clarified more clearly. For example, what is the purpose of constraint identification and decomposition, and why can it solve the problems mentioned in the introduction.**
>
> Thank you for your thoughtful feedback. The goal of READ-SQL is to bridge the gap between natural language and SQL within an end-to-end generation framework. We emphasize establishing connections between the constraints in the problem and the corresponding SQL clauses.  We highlight the key steps of our READ-SQL to improve performance as follows:
>
> - Enhancing the model's understanding of the nuances in SQL questions by incorporating additional question/SQL pairs for training;
> - Providing `reason` to bridge the gap between the question and the SQL statement;
> - Refining low-confidence clauses to further minimize the occurrence of erroneous constraints.
>
> The implementation of these solutions relies on READER, which identifies and decomposes constraints, generates sub-SQL statements, and establishes reasoning paths.
>
> We specifically examine the performance of READ-SQL and SFT CodeS in addressing the three types of errors mentioned in the introduction. The results are detailed in the revised paper (lines 514-521). READ-SQL effectively reduces the occurrence of these errors, particularly the phenomenon of omitted constraints, which has been decreased by nearly 48.7%.

---

> ### Author Response · Authors · 2024-11-25
> **Response to Reviewer GSF2 (Part 2)**
>
> > **W4: Some technical introductions are unclear, which makes it difficult to understand the paper.**
>
> We provide detailed responses to your questions:
>
> ---
>
> > **W4-1:** What does the constraint or sub-SQL mean, and what can be considered as a constraint?
>
> **Constraint:** In simple terms, a constraint refers to each operation expression in an SQL query.  For example, in the SQL query `"SELECT name FROM person WHERE age=18"`, components like `"SELECT name"`,` "FROM person"`, and `"WHERE age =18"` are all considered constraints. A detailed definition of constraints can be found in Appendix A.1 (the overall process of READER) and Figure 16 (constraints identified in the AST).
>
> **sub-SQL:**  A sub-SQL is an SQL query derived by removing one or more constraints from the original SQL query. For instance, in the above SQL, removing the constraint `"WHERE age=18"` results in `"SELECT name FROM person"`, which is a sub-SQL. A more complex example is shown in Figure 8 (lines 2050).
>
> ---
>
> > **W4-2:** What are the goals of the Generator and Corrector, how are they used in SQL generation, and how is the Corrector trained and realized?
>
> - **The goal of the Generator** is to minimize problematic constraints--specifically misinterpreted, omitted, and unwanted constraints--as illustrated in Fig. 1. This is achieved through a well-designed mechanism that effectively generates augmented data for further training.
> - **The goal of the Corrector** is to refine low-confidence constraints in the Generator and reduce the errors of the Generator.
>
> The reasoning process of the Generator and the Corrector is as addressed in A3 of our Global Comment. Secondly, the training of the Coeector includes the following steps:
>
> - **Construct the training set of the Corrector:** We perform four-fold cross-validation on the BIRD and Spider training sets, respectively. We train the Generator on any three folds and perform reasoning on the remaining fold. According to the reasoning results, the training data of each fold is constructed, and finally combined to form the training sets of the Corrector on BIRD and Spider, respectively; see more details in Appendix A.4 (lines 1101-1108).
> - **Fine-tuning training:** The training of the Corrector also uses LoRA fine-tuning with the default setting.
>
> ---
>
> > **W4-3:** The proposed method regenerated the question based on the sub-SQL, so what is the relation between the generated question and the original ones, and have the authors considered their consistency? Besides, how did the authors ensure that the sub-SQL could solve the generated question?
>
> Figure 10(lines 2105) illustrates an example of the original (question, SQL) pair alongside two (sub-question, sub-SQL) pairs. It demonstrates that the generated questions differ from the original primarily due to difference in only one constraint.
>
> To address the consistency issue, as shown in Figure 9(lines 2080), we utilize the original question pair as a demonstration for the LLM, guiding it to generate a question that corresponds to the sub-SQL. Our aim is to closely mimic the format of the original query to ensure consistency between the two. Additionally, we select high-quality generated data through a combination of rule-based criteria and manual screening, as detailed in lines 1023-1079 of the paper.

---

> > ### Comment · Reviewer_GSF2 · 2024-11-26
> >
> > Thanks a lot for your efforts on the responses. However, the responses do not fully address my concerns. For example, for W4-2, it is still unclear how the training data of Corrector matches the inference. For W4-3, it is doubtful whether taking the orginal question as a demonstration could make the LLM generate a question with high consistency to the original question. It is also unclear whether the sub-SQL could answer the generated question. More importantly, the technical contribution of the paper appeals less compelling. Therefore, I would keep my scores.

---

### Official Review · Reviewer_HCL3 · 2024-10-30

**Soundness:** 2
**Presentation:** 3
**Contribution:** 2
**Rating:** 3
**Confidence:** 5

**Summary:**

READ-SQL is a framework designed to improve the conversion of natural language queries into SQL commands, an area known as text-to-SQL. This framework incorporates two main models, Generator and Corrector, which work with a core module, READER. READER decomposes SQL queries into components, helping to build reasoning paths. The Generator initially formulates SQL based on input questions and generated data, while the Corrector refines these SQLs by focusing on high- and low-confidence components. Together, they achieve significant accuracy and efficiency improvements over comparable models, even those with larger parameter sizes, as shown in the extensive experiments on benchmarks like BIRD and Spider​.

**Strengths:**

1. READ-SQL outperforms existing models on the BIRD benchmark, achieving high execution accuracy and efficiency.
2. Extensive testing on multiple datasets and with various experimental setups (e.g., ablation studies, different model sizes) demonstrates the framework's consistency and robustness.

**Weaknesses:**

1. The paper highlights three key issues in text-to-SQL: misinterpreted constraints, omitted constraints, and unwanted constraints. However, these issues are neither addressed nor mentioned in the experimental section, nor are they resolved in the proposed approach, which I find confusing.
2. The authors state, “However, existing methods primarily focus on optimizing SQL generation without improving the model’s understanding of the question or establishing strong relationships between questions and SQL clauses.” From my perspective, schema linking already serves to establish relationships between queries and SQL structure, which weakens this point as a unique limitation. Additionally, READ-SQL does not appear to be significantly better than other methods in terms of understanding the question. Furthermore, they mention, “Additionally, since these methods do not employ an end-to-end architecture, there is potential for information loss during the process.” Based on my understanding, T5-based and GPT-4-based approaches share a similar end-to-end structure with READ-SQL, as they transform inputs directly into SQL queries. Therefore, this limitation also seems unconvincing. Overall, the motivation for READ-SQL remains unclear to me.
3. Although augmented data enhances performance, generating question/reason pairs involves complex processing that may pose implementation challenges. More importantly, even with additional training data, the performance gains on the Spider dataset are less significant than on the BIRD dataset. This raises the question of whether the observed improvements might be due to overfitting, as the paper indicates that most of the augmented data originates from the BIRD dataset.

**Questions:**

1. What specific advancements does READER bring compared to earlier SQL decomposers?
2. How scalable is READ-SQL beyond 3 billion parameters, and are there plans for optimizing even larger-scale models?
3. Through augmented data generation, where question/SQL pairs and question/reason pairs are created, READ-SQL enhances comprehension of complex queries. Open-sourcing this related data would further support the broader community.

---

> ### Author Response · Authors · 2024-11-25
> **Response to Reviewer HCL3 (Part 1)**
>
> Thank you for your thoughtful feedback and thorough review. Below, we provide responses to your questions and concerns.
> > **W1: The paper highlights three key issues in text-to-SQL: misinterpreted constraints, omitted constraints, and unwanted constraints. However, these issues are neither addressed nor mentioned in the experimental section, nor are they resolved in the proposed approach, which I find confusing.**
>
> Thank you for your thoughtful feedback.  We provide a more detailed analysis in the experimental section. Additionally, we emphasize that the goal of READ-SQL is to enhance the model's sensitivity to constraints in the question and to regenerate low-confidence constraints, thereby reducing the occurrence of these three issues. In the revised paper, lines 514-521, We add a relevant experiment to confirm this conclusion:
>
> |                                                   | Unwanted constraints | Misinterpreted constraints | Omitted constraints |
> | ------------------------------------------------- | -------------------- | -------------------------- | ------------------- |
> | READ-SQL 3B is correct, but SFT CodeS 3B is wrong | 25                   | 52                         | 41                  |
> | READ-SQL 3B is wrong, but SFT CodeS 3B is correct | 18(↓ 28%)            | 43(↓ 17.3%)                | 21(↓ 48.7%)         |
>
> We present the frequency of the three types of errors occurring in two scenarios: cases where READ-SQL is  correct but SFT-CodeS is wrong, and vice versa. It can be seen that READ-SQL has reduced the occurrence of all three error types, particularly the omission errors, which decreased by 48.7%.
>
>
> ---
>
> > **W2: The authors state, “However, existing methods primarily focus on optimizing SQL generation without improving the model’s understanding of the question or establishing strong relationships between questions and SQL clauses.” From my perspective, schema linking already serves to establish relationships between queries and SQL structure, which weakens this point as a unique limitation. ... ...Therefore, this limitation also seems unconvincing. Overall, the motivation for READ-SQL remains unclear to me.**
>
> We answer the questions one-by-one below:
>
> > **W2-1 :** schema linking already serves to establish relationships between queries and SQL structure.
>
> We highlights that while pattern linking helps the model establish connections between the problem and SQL, we still observe instances of misinterpreted constraints, omitted constraints, and unwanted constraints in the model's output under the schema linking scheme proposed by CodeS, a state-of-the-art method, the results in Appendix A.5.2 of the revised paper.  Therefore, we hypothesize that these critical errors in SQL generation arise from the structural differences between natural language and SQL.  Natural language is often vague and changeable, whereas SQL is structured programming language.  Our research focuses on how to bridge this gap.
>
> > **W2-2:** READ-SQL does not appear to be significantly better than other methods in terms of understanding the question.
>
> We specifically explore the performance of READ-SQL and SFT CodeS in the face of the three types of errors mentioned in the introduction. The results are shown in the revised paper at lines 514-521. READ-SQL effectively reduces the occurrence of the three situations, especially the occurrence of the omitted constraint phenomenon, which is reduced by nearly 48.7%.
>
> > **W2-3:**   The motivation for READ-SQL is unclear
>
> Just as W2-1 replied, bridging the gap between natural language and SQL is the focus of our work. In the Introduction section (lines 070-081), we outline two types of methods that use intermediate representations to address these differences: (1) SQL-like grammar languages and (2) direct output of SQL structures. However, these two types of methods do not support end-to-end SQL generation, which may result in the loss of critical information."
>
> Differently, our READ-SQL provides an end-to-end generation solution.  We emphasize establishing connections between the constraints in the problem and the corresponding SQL clauses. We highlight the key steps of our READ-SQL to improve performance as follows:
>
> - Enhancing the model's understanding of the nuances in SQL questions by incorporating additional question/SQL pairs for training;
> - Providing `reason` to bridge the gap between the question and the SQL statement;
> - Refining low-confidence clauses to further minimize the occurrence of erroneous constraints.
>
> The above three points need to be achieved with the help of our invention, READER, the SQL decomposer. Furthermore, our focus is not solely on the end-to-end generation of SQL, but on establishing the connection between the question and SQL within an end-to-end framework to bridge the gap between natural language and SQL.

---

> ### Author Response · Authors · 2024-11-25
> **Response to Reviewer HCL3 (Part 2)**
>
> > **W3: Although augmented data enhances performance, generating question/reason pairs involves complex processing that may pose implementation challenges. More importantly, even with additional training data, the performance gains on the Spider dataset are less significant than on the BIRD dataset. This raises the question of whether the observed improvements might be due to overfitting, as the paper indicates that most of the augmented data originates from the BIRD dataset.**
>
> Thank you for your insightful feedback. We respond to your questions as follows:
>
> ---
>
> > **W3-1:** There is a complex process involved in generating question/reason pairs.
>
>  **A3-1:**  We emphasize that by executing READER on a SQL query, it can easily derive the corresponding clauses, sub-SQLs, and reasoning paths. We then use the LLM to generate (sub-question, sub-SQLs) and (question, reason) pairs, where reason is a description of the reasoning path of constructing the sub-SQLs.  This procedure can be further explored by simply rule-based methods.
>
> ---
>
> > **W3-2:  **Whether the observed improvements might be due to overfitting?
>
>  **A3-2:**   We conduct a difficulty analysis and comparison between READ-SQL and SFT CodeS on three Spider variants. The results show that READ-SQL is ahead of SFT CodeS in EX indicators. To further explore the model's generalization, we have provided more detailed results in A2 of the Global Comment.
>
> ---
>
> > **W3-3:** The most of the augmented data originates from the BIRD dataset.
>
>  **A3-3:**  Our enhanced data will be generated not only for the BIRD training set but also for the Spider training set.
>
> ---
>
> > **Q1: What specific advancements does READER bring compared to earlier SQL decomposers?**
>
> READER is a self-developed method based on SQLglot. Compared with the existing SQL decomposition, it can divide SQL clauses more finely, split all possible executable sub-SQLs, and obtain multiple reasoning paths. We compare it with two recently published works, for example:
>
> - STEPS\[1\]: It cannot fine-grain SQL, resulting in clauses not being independent units.
> - DeSQL\[2\]: The SQL decomposer used has limited SQL structures involved, cannot handle nested queries, etc., and can only decompose simple SQL.
>
> The following are detailed comparison results:
>
> |        | Split Clause    | Synthetic sub-SQL | Multiple reasoning paths | Support for complex syntax |
> | ------ | --------------- | ---------------- | ------------------------ | -------------------------- |
> | READER | ✓（Refinement） | ✓                | ✓                        | ✓                          |
> | STEPS  | ✓               | ✗                | ✗                        | ✓                          |
> | DeSQL  | ✓（Refinement） | ✓                | ✗                        | ✗                          |
>
> We show examples of these three methods in Appendix A.1.4 of the revised paper.
>
> \[1\] Yuan Tian, Zheng Zhang, Zheng Ning, Toby Jia-Jun Li, Jonathan K. Kummerfeld, and Tianyi Zhang. Interactive text-to-sql generation via editable step-by-step explanations. In EMNLP, pp. 16149–16166. Association for Computational Linguistics, 2023.
>
> \[2\] Sabaat Haroon, Chris Brown, and Muhammad Ali Gulzar. Desql: Interactive debugging of SQL in data-intensive scalable computing. Proc. ACM Softw. Eng., 1(FSE):767–788, 2024
>
> ---
>
> > **Q2: How scalable is READ-SQL beyond 3 billion parameters, and are there plans for optimizing even larger-scale models?**
>
> READ-SQL can certainly work on larger-scale models. The reason for this is that we expand our training datasets with new question/SQL and question/reason pairs. The new question/SQL pairs enhance the model's sensitivity to questions, while the question/reason pairs improve its reasoning ability in end-to-end generation. Additionally, we use READER to enable the Corrector to re-evaluate low-confidence clauses, further boosting text-to-SQL accuracy.
>
> However, due to our current resource limitations, we can only conduct experiments with models up to 3B. Nonetheless, we are confident that READ-SQL can scale effectively and deliver even better performance when applied to models with more parameters and augmented data.
>
> ---
>
> > **Q3: Through augmented data generation, where question/SQL pairs and question/reason pairs are created, READ-SQL enhances comprehension of complex queries. Open-sourcing this related data would further support the broader community.**
>
> We commit to following ICLR's reproducibility guidelines and to making data, code, and runtime logs public.

---

> > ### Comment · Reviewer_HCL3 · 2024-11-25
> > **Thanks for the authors' efforts.**
> >
> > First of all, I would like to thank the authors for their efforts in addressing my questions. While the proposed method shows some relative improvement, its novelty appears less compelling compared to existing works such as codeS, particularly in addressing my concerns regarding W2 (schema linking and other end-to-end methods). So, I will keep my score.

---

### Official Review · Reviewer_ezyC · 2024-11-02

**Soundness:** 2
**Presentation:** 3
**Contribution:** 2
**Rating:** 5
**Confidence:** 4

**Summary:**

READ-SQL introduces an innovative approach that leverages Abstract Syntax Trees (ASTs) to efficiently extract chains of reasoning by incorporating each SQL constraint, as well as to identify all sub-SQL components. This strategy supports their development of a two-step process involving generation and correction. The generator is trained on both SQL generation and reasoning tasks, enabling the model to learn step-by-step reasoning necessary for producing accurate SQL queries. Using the constraints generated from the READER, corrector will find the low-confidence constraints and try to fix them. Experimental results indicate that this method achieves state-of-the-art (SOTA) performance with open-source LLMs under 3 billion parameters.

**Strengths:**

1) The primary strength of this paper lies in the READER method. By decomposing SQL queries into sub-queries, the proposed approach effectively facilitates multi-step reasoning, guiding the model to generate the final answer.

2) The proposed approach achieves state-of-the-art performance compared to all models with fewer than 3 billion parameters on two well-known benchmarks: BIRD and Spider.

**Weaknesses:**

1) The main weakness of this paper lies in the experimental section. The baselines used are primarily methodologies that do not represent the most recent advancements in the field, such as Distillery, CHASE-SQL, and CHESS. Although the paper states that the main baseline is the SFT CodeS model itself, it would be beneficial to compare their approach with more advanced fine-tuning methods, such as using zero-shot chain-of-thought prompting to generate reasoning steps and training the CodeS model with those reasoning paths.

2) According to the results shown in Table 2, there are two instances where the performance of READ-SQL is lower than the SFT version of the CodeS model. This may indicate overfitting on the Spider dataset, which could explain why READ-SQL outperforms the SFT CodeS model on Spider specifically.

3) To better assess the effectiveness of their proposed correction methodology, it would be valuable to compare it with basic self-correction methods, such as those outlined in the DIN-SQL paper. This comparison would help determine whether the improvement comes solely from an additional LLM call or from the specific advantages of the proposed method.

**Questions:**

N/A

---

> ### Author Response · Authors · 2024-11-25
> **Response to Reviewer ezyC**
>
> Thank you for your thoughtful feedback and thorough review. Below, we provide responses to your questions and concerns.
> > **W1: The main weakness of this paper lies in the experimental section. The baselines used are primarily methodologies that do not represent the most recent advancements in the field, such as Distillery, CHASE-SQL, and CHESS. Although the paper states that the main baseline is the SFT CodeS model itself, it would be beneficial to compare their approach with more advanced fine-tuning methods, such as using zero-shot chain-of-thought prompting to generate reasoning steps and training the CodeS model with those reasoning paths.**
>
> Thank you for your valuable feedback. In the revised paper, we have incorporated comparisons with the current state-of-the-art benchmarks, as shown in Table 1 and Table 2, while these benchmarks predominantly rely on closed-source models. We emphasize that our work focuses on small-scale models and their applicability in resource-constrained environments.  In addition, we also experiment with the fine-tuning method you mentioned, using zero-shot chain-of-thought to generate reasoning. Below is a comparison of its performance with the reasoning path method we obtained through READER parsing:
>
> | Question      | Name movie titles released in year 1945. Sort the listing by the descending order of movie popularity. |
> | ------------- | ------------------------------------------------------------ |
> | Evidence      | released in the year 1945 refers to movie_release_year = 1945; |
> | SQL           | SELECT movie_title FROM movies WHERE movie_release_year = 1945 ORDER BY movie_popularity DESC LIMIT 1 |
> | Our method    | First, gather all movie data, then sort by descending popularity, filter to include only 1945 releases, **limit the results to the top movie**, and finally, select only the movie title. |
> | Zero-shot CoT | Select movie titles from the movies table, filter for those released in 1945, sort by descending popularity, list the titles. |
>
> Our method ensures that no SQL clauses are omitted, unlike zero-shot CoT, which may fail to retain clauses such as `LIMIT 1`. The prompts for zero-shot CoT are included in the supplementary materials, while the prompt used in our method is presented in Figure 12 of our paper.
>
> ---
>
> > **W2: According to the results shown in Table 2, there are two instances where the performance of READ-SQL is lower than the SFT version of the CodeS model. This may indicate overfitting on the Spider dataset, which could explain why READ-SQL outperforms the SFT CodeS model on Spider specifically.**
>
> Thank you for your insightful observation.  We conduct a difficulty analysis and comparison between READ-SQL and SFT CodeS on three Spider variants; see Appendix A.5.9 (lines 1323-1348) in the revision paper.  The results show that READ-SQL is ahead of SFT CodeS in EX indicators. To further explore the model's generalization, we have provided more detailed explanations in A2 of the Global Comment.
>
>
>
> ---
>
> > **W3: To better assess the effectiveness of their proposed correction methodology, it would be valuable to compare it with basic self-correction methods, such as those outlined in the DIN-SQL paper. This comparison would help determine whether the improvement comes solely from an additional LLM call or from the specific advantages of the proposed method.**
>
> We present the comparison between standard self-correction (GPT-4o) and Corrector (GPT-4o) in Table 4. The results confirm that the improvement in Corrector's capabilities is not solely attributed to GPT-4o.
>
> In addition, we compare Corrector (GPT-4o) with the self-correction method in DIN-SQL, and the results are as follows:
>
> |                                | BIRD Dev EX | API tokens (input) |
> | ------------------------------ | ----------- | ------------------ |
> | Generator-3B                   | 56.98       | \                  |
> | Generator-3B+DIN-SQL(GPT-4o)   | 55.93       | 4501               |
> | Generator-3B+Corrector(GPT-4o) | 61.21       | 2619               |
>
> Notably, DIN-SQL's self-correction method may lead to additional reasoning outputs, resulting in higher token consumption.  The results demonstrate that our self-correction method significantly improves performance while incurring less API overhead. We show the prompts used by DIN-SQL and Corrector (GPT-4o) in Figures 13 and 14 of the revised paper.

---

### Official Review · Reviewer_8arY · 2024-11-04

**Soundness:** 3
**Presentation:** 3
**Contribution:** 3
**Rating:** 6
**Confidence:** 4

**Summary:**

This paper introduces READ-SQL, a novel framework for enhancing the text-to-SQL task by using a reasoning path decomposer called READER. The framework aims to address challenges such as misinterpreted or omitted constraints in SQL generation. READ-SQL comprises a Generator and a Corrector, both of which leverage READER to break down SQL queries into sub-SQLs and reasoning paths for better data preparation and correction. Extensive experiments show that READ-SQL outperforms competing baselines, achieving strong results with fewer parameters and demonstrating applicability across different models and scenarios.

**Strengths:**

1. The proposed framework achieves promising results using small LLMs (1B and 3B models), which is advantageous for local deployment and reduces computational resource requirements in real-world applications. Given that the text-to-SQL community primarily focuses on larger proprietary LLMs, this work advances research on smaller-scale LLMs, providing practical benefits for scenarios such as on-device AI.

2. This paper introduces common errors in text-to-SQL and presents a detailed error analysis to validate the framework. Experiments conducted on five well-recognized datasets, including BIRD and Spider variants, provide strong evidence of the framework's effectiveness, showcasing comprehensive experimental support.

3. Overall, the framework is technically sound and novel. The paper is well-structured and easy to follow, effectively presenting the methodology, experiments, and results, which allows readers to easily understand the contributions and significance of the proposed approach.

**Weaknesses:**

1. The baseline included in the paper is insufficient. As a submission in October 2024, and in reference to recent studies [1][2], the paper should compare against more advanced baselines, such as [3][4][5][6]. Although this comparison of performance may not be entirely fair due to the varying scale of model parameters, the authors could highlight their advantages in other aspects, such as model size, GPU resources, and runtime. Comparing to up-to-date baselines would strengthen the paper's position for submission to a top conference.

2. The paper lacks a detailed runtime analysis. Given the focus on using smaller-scale LLMs, adding an analysis of training and inference times would underscore the practical advantages of the lightweight design. Such an analysis would further emphasize the framework's efficiency in real-world settings, particularly for on-device and resource-constrained applications, thereby enhancing the paper's impact.

3. The performance of the proposed framework seems limited in terms of generalization across different datasets. While it shows significant improvement on the more challenging BIRD dataset (with more complex database environments), its performance on simpler Spider-based databases is less pronounced, and in some cases, even demonstrates negative optimization. This discrepancy seems counterintuitive, especially since the framework does not explicitly claim to be suited only for complex scenarios.

[1] Zijin Hong, et al. "Next-Generation Database Interfaces: A Survey of LLM-based Text-to-SQL" arXiv preprint, 2024.

[2] Xinyu Liu, et al. "A Survey of NL2SQL with Large Language Models: Where are we, and where are we going?" arXiv preprint, 2024.

[3] Tonghui Ren, et al. "PURPLE: Making a Large Language Model a Better SQL Writer" In Proceedings of ICDE, 2024.

[4] Mohammadreza Pourreza, et al. "DTS-SQL: Decomposed Text-to-SQL with Small Large Language Models" arXiv preprint, 2024

[5] Boyan Li, et al. "The Dawn of Natural Language to SQL: Are We Fully Ready?" In Proceedings of VLDB, 2024.

[6] Ge Qu, et al. "Before Generation, Align it! A Novel and Effective Strategy for Mitigating Hallucinations in Text-to-SQL Generation" In Findings of ACL, 2024.

**Questions:**

1. Can READ-SQL work on larger-scale models (e.g., 13B, 34B LLMs)? Since the proposed framework already achieves comparable performance with small-scale LLMs, it would be interesting to explore whether the framework could outperform GPT-4-based methods when incorporating 70B models. Alternatively, if READ-SQL cannot work on larger-scale models, it would be helpful to explain why.

2. Since the most direct comparison for the model at the same scale is with SFT CodeS-1B and SFT CodeS-3B, could the authors provide a detailed analysis of the specific types of questions where READ-SQL shows improvement? A statistical analysis of correct samples based on difficulty level, along with a straightforward case study comparing READ-SQL with CodeS, would help readers better understand the contributions and improvements.

3. Does READ-SQL struggle when handling ambiguous questions? The improvement brought by READ-SQL in robustness datasets such as Spider-Syn and Spider-Realistic (+1.3 and +0.4) is not as significant as in standard datasets like BIRD and Spider (+2.35 and +0.8). Moreover, why does READ-SQL-3B even show negative optimization in Spider-DK (-0.2)?

---

> ### Author Response · Authors · 2024-11-25
> **Response to Reviewer 8arY (Part 1)**
>
> Thank you for your thoughtful feedback and thorough review. Below, we provide responses to your questions and concerns.
>
> > **W1: The baseline included in the paper is insufficient. As a submission in October 2024, and in reference to recent studies \[1\]\[2\], the paper should compare against more advanced baselines, such as \[3\]\[4\]\[5\]\[6\]. Although this comparison of performance may not be entirely fair due to the varying scale of model parameters, the authors could highlight their advantages in other aspects, such as model size, GPU resources, and runtime. Comparing to up-to-date baselines would strengthen the paper's position for submission to a top conference.**
>
> Thank you for your valuable feedback. In the revised paper, we have incorporated comparisons with the current state-of-the-art benchmarks,as shown in Table 1 and Table 2, while these benchmarks predominantly rely on closed-source models. We emphasize that our work focuses on small-scale models and their applicability in resource-constrained environments. However, due to limited resources, we are unable to fully reproduce the larger open source model as well as the closed source model to gain running time and API overhead. We highlight model size as a metric in the revised paper.
>
>
>
> ---
>
> > **W2: The paper lacks a detailed runtime analysis. Given the focus on using smaller-scale LLMs, adding an analysis of training and inference times would underscore the practical advantages of the lightweight design. Such an analysis would further emphasize the framework's efficiency in real-world settings, particularly for on-device and resource-constrained applications, thereby enhancing the paper's impact.**
>
> Thank you for valuable suggestions.   In Appendix A.5.5 of the revised paper, we have updated it as follows:
>
> We conduct a comparative runtime analysis of READ-SQL and SFT-CodeS on NVIDIA RTX 3090 GPUs with 24 GB of memory.  The experiments are carried out using a two-card setup for training and a single-card setup for inference, the result as follows:
>
>
> | Model        | Model size | Training time   | Estimated memory usage (only load model) | Inference time (s/sample) | BIRD Dev(EX)  |
> | :----------- | :--------- | :-------------- | :--------------------- | :------------------------ | :------------ |
> | SFT CodeS-3B | 3B         | out of memory   | 6G                     | 2.81                      | 55.02         |
> | SFT CodeS-7B | 7B         | out of memory   | 14G                    | 6.51                      | 57            |
> | READ-SQL-3B  | 3B         | 8h 23m + 3h 24m | 6.1G                   | 4.65(2.97+1.68)           | (56.98/57.37) |
>
> Among them, SFT CodeS adopts full fine-tuning, and the training time cannot be displayed due to resource constraints.  READ-SQL comprises two components: the Generator and the Corrector.  Data is presented in a sequence where the Generator precedes the Corrector.
> We get the following conclusion:
> - READ-SQL offers an advantage of achieving high performance with minimal memory usage.
> - Leveraging the LoRA fine-tuning method, it enables efficient training and inference in low-resource environments compared to full fine-tuning.
> - Using only the Generator-3B improves performance by 1.96\% compared to a single 3B model (SFT CodeS-3B), with slight increase in inference time (2.97s vs. 2.81s) due to additional computation on the LoRA.  Remarkably, its performance is only 0.02% lower than that of SFT CodeS-7B.
> - READ-SQL-3B surpasses SFT CodeS-7B by nearly 0.37\% with smaller inference time and less memory usage.
>
>
> ---
>
> > **W3&Q3:  The model's generalization and its improvements on Spider datasets and their variants are not as strong as those observed with the BIRD dataset.**
>
> Thank you for your insightful questions. We conduct a difficulty analysis and comparison between READ-SQL and SFT CodeS on three spider variants. The results show that READ-SQL is ahead of SFT CodeS in EX indicators. To further explore the model's generalization, we have provided more detailed explanations in A2 of the Global Comment.

---

> ### Author Response · Authors · 2024-11-25
> **Response to Reviewer 8arY (Part 2)**
>
> > **Q1: Can READ-SQL work on larger-scale models (e.g., 13B, 34B LLMs)? Since the proposed framework already achieves comparable performance with small-scale LLMs, it would be interesting to explore whether the framework could outperform GPT-4-based methods when incorporating 70B models. Alternatively, if READ-SQL cannot work on larger-scale models, it would be helpful to explain why.**
>
> READ-SQL can certainly work on larger-scale models. The reason for this is that we expand our training datasets with new question/SQL and question/reason pairs. The new question/SQL pairs enhance the model's sensitivity to questions, while the question/reason pairs improve its reasoning ability in end-to-end generation. Additionally, we apply READER to enable the Corrector to re-evaluate low-confidence clauses, further boosting text-to-SQL accuracy.
>
> However, due to our current resource limitations, we have not been able to experiment with these larger-scale models. Nonetheless, we are confident that READ-SQL can scale effectively and deliver even better performance when applied to models with more parameters.
>
> ---
>
>
> > **Q2: Since the most direct comparison for the model at the same scale is with SFT CodeS-1B and SFT CodeS-3B, could the authors provide a detailed analysis of the specific types of questions where READ-SQL shows improvement? A statistical analysis of correct samples based on difficulty level, along with a straightforward case study comparing READ-SQL with CodeS, would help readers better understand the contributions and improvements.**
>
> Thank you for your thoughtful suggestion. We conduct a statistical analysis of the BIRD Dev, Spider Dev and Spider variants based on difficulty level (the results are added to the revised paper), as shown in Appendix A.5.9(lines 1323-1348). The following are the results on BIRD Dev:
>
> | Model        | Simple       | Moderate     | Challenging  | Total        |
> | :----------- | :----------- | :----------- | :----------- | :----------- |
> | SFT CodeS-1B | 57.95        | 37.42        | 34.72        | 49.54        |
> | Generator-1B | 59.89(+1.94) | 41.29(+3.87) | 33.33(-1.39) | 51.76(+2.22) |
> | READ-SQL-1B  | 60.56(+2.61) | 43.87(+6.45) | 31.94(-2.78) | 52.87(+3.33) |
> | SFT CodeS-3B | 63.35        | 44.30        | 36.11        | 55.02        |
> | Generator-3B | 65.08(+1.73) | 46.67(+2.37) | 38.19(+2.08) | 56.98(+1.96) |
> | READ-SQL-3B  | 65.41(+2.06) | 47.53(+3.23) | 37.50(+1.39) | 57.37(+2.35) |
>
> Our findings in BIRD are as follows:
>
> - The Generator generally outperforms the SFT-CodeS, showing the greatest improvement at moderate difficulty levels.
> - With the introduction of the Corrector, there are improvements at both simple and moderate difficulty levels, though its effectiveness may decrease at more challenging difficulty levels.
>
> The following are the results on Spider Dev:
>
> | Model        | Easy         | Medium       | Hard         | Extra        | All          |
> |:-------------|:-------------|:-------------|:-------------|:-------------|:-------------|
> | SFT CodeS-1B | 91.10        | 83.60        | 68.40        | 51.80        | 77.80        |
> | Generator-1B | 91.90(+0.80) | 83.60(+0.00) | 73.60(+5.20) | 53.00(+1.20) | 80.20(+2.40) |
> | READ-SQL-1B  | 91.90(+0.80) | 86.80(+3.20) | 74.70(+6.30) | 53.60(+1.80) | 80.70(+2.90) |
> | SFT CodeS-3B | 93.50        | 87.00        | 73.60        | 61.40        | 82.20        |
> | Generator-3B | 94.80(+1.30) | 88.60(+1.60) | 79.90(+6.30) | 58.40(-3.00) | 83.80(+1.60) |
> | READ-SQL-3B  | 94.80(+1.30) | 89.20(+2.20) | 81.00(+7.40) | 58.40(-3.00) | 84.20(+2.00) |
>
> Our findings in Spider Dev are as follows:
>
> - The Generator exhibits the highest growth rate at hard difficulty.
> - The addition of the Corrector does not lead to significant improvements. We believe this is because the Spider dataset, unlike the BIRD dataset, lacks additional external information. As a result, Corrector relies solely on column names to determine clause constraints.
>
> In addition, we provide a case study with multiple samples showcasing a direct comparison between READ-SQL and SFT CodeS in revised paper as shown in Table 19 to 24 (lines 1366-1457).

---

> > ### Comment · Reviewer_8arY · 2024-11-26
> >
> > Thank the authors for their detailed response. Overall, I believe this paper meets the acceptance threshold. However, as mentioned in my earlier feedback, I think the proposed method has some limitations in terms of performance and scalability on specific datasets. Therefore, I am inclined to keep my rating.

---

### Official Review · Reviewer_E8Xr · 2024-11-06

**Soundness:** 3
**Presentation:** 2
**Contribution:** 3
**Rating:** 5
**Confidence:** 4

**Summary:**

This paper presents READ-SQL, a framework that uses SQL decomposition for two main purposes: (1) data augmentation for training an SQL generator and (2) self-correction by training a corrector that uses common and differing clauses among multiple generated SQLs to refine the SQLs. The model demonstrates improved performance over baselines on the BIRD, Spider, and various Spider benchmark variants.

Some comments on presentation and typos:

1. Line 301: "the above three kinds of data" — only two types are explicitly listed.

2. Figure 4: The caption refers to a yellow color not present in the figure.

3. Table 4: The first two rows label “coder-1.3B-1.3B” with “1.3B” duplicated unnecessarily.

**Strengths:**

1. The proposed SQL decomposer, utilizing AST parsing, is both novel and efficient, providing a scalable approach.

2. The self-correction method based on common and differing clauses is innovative.

3. The framework enhances performance for smaller models.

**Weaknesses:**

1. The writing could be improved to be more direct and accessible. Section 3.4 is incomplete — it only describes the extraction of common and different constraints without explaining the subsequent self-correction process. Some insights can be found in Figure 4, but the absence in the main text hinders comprehension.

2. The comparisons with other models are somewhat unfair; READ-SQL-3B includes two 3B models (i.e., a 3B generator and a 3B corrector), so a direct comparison with CodeS-3B may not be appropriate.

3. The performance improvements are modest, especially considering the combined use of two 3B models.

**Questions:**

1. Line 44 mentions three typical categories of text-to-SQL errors. Are there any statistics on the frequency of these error types?

2. How would the performance be affected if question-SQL pairs and question-reason pairs were combined into question-reason-SQL chain-of-thought data for fine-tuning CodeS?

---

> ### Author Response · Authors · 2024-11-25
> **Response to Reviewer E8Xr**
>
> Thank you for your careful observation. We have corrected the presentation and spelling errors, as shown in the revised paper. Below, we address your questions and concerns.
>
> > **W1: The writing could be improved to be more direct and accessible. Section 3.4 is incomplete — it only describes the extraction of common and different constraints without explaining the subsequent self-correction process. Some insights can be found in Figure 4, but the absence in the main text hinders comprehension.**
>
> Thank you for your valuable suggestion. We have incorporated your feedback into the revision and updated it in Sec. 3.4 of the reviewed version.  For more details, please refer to A3 of the Global Comment.
>
> ---
>
> > **W2: The comparisons with other models are somewhat unfair; READ-SQL-3B includes two 3B models (i.e., a 3B Generator and a 3B corrector), so a direct comparison with CodeS-3B may not be appropriate.
> > &W3: The performance improvements are modest, especially considering the combined use of two 3B models.**
>
> We fine-tune CodeS-3B using LoRA to create a Generator adapter and a Corrector adapter, resulting in a minimal parameter increase of only 0.972% (0.486% per adapter), far from constituting two separate 3B models. Notably, READ-SQL surpasses the SFT CodeS-7B SOTA model on BIRD Dev, showcasing substantial improvement at the 3B scale. For a fair comparison, the results of individual Generators are detailed in A2 of the Global Comment.
>
>
> ---
> > **Q1: Line 44 mentions three typical categories of text-to-SQL errors. Are there any statistics on the frequency of these error types?**
>
> In the revised paper, lines 1146-1150, we have updated it accordingly.  We randomly select 100 error samples from the 690 total errors in SFT CodeS-3B and manually categorized them (Note: a single erroneous sample may contain multiple errors).  The results are shown as follows:
>
> |            | Unwanted constraint | Misinterpreted constraint | Omitted constraint | Error gold |
> | ---------- | ------------------- | ------------------------- | ------------------ | ---------- |
> | Percentage | 13%                 | 47%                       | 43%                | 7%         |
>
> ---
>
> > **Q2: How would the performance be affected if question-SQL pairs and question-reason pairs were combined into question-reason-SQL chain-of-thought data for fine-tuning CodeS ?**
>
> Thank you for your interesting suggestion.  We have added additional experiments in lines 1206-1217 in the revised paper.  We apply the **Reason-SQL Formatting (RSF)** method, as mentioned, setting the output format to REASON: `{reason text};\nSQL: {SQL query}`. We train on the BIRD training set and reason on BIRD Dev.  The results are as follows:
>
> |      | RSF-1B | Generator-1B | RSF-3B | Generator-3B |
> | ---- | ------ | ------------ | ------ | ------------ |
> | EX   | 40.94  | 51.76        | 48.63  | 56.98        |
>
> This approach is clearly unsuitable for small language models (SLMs) for the following reasons:
>
> - The `reason` output during model inference contributes to error accumulation, interfering with subsequent decoding and introducing unnecessary errors.
> -  Beam search is used for decoding, and including the `reason` output reduces the available space for the SQL portion, limiting the model's ability to generate diverse SQL queries.

---

### Author Response · Authors · 2024-11-25
**Global Comment (part 1)**

> **Q1: Contribution of our Work**

**A1:** We argue that the contribution of READ-SQL is not merely an incremental improvement over existing Text-to-SQL methods; rather, it has the potential to inspire mainstream adoption for real-world Text-to-SQL applications.

Additionally, the developed SQL decomposer, READER, has proven effective in enhancing performance, with a 3B LoRA model surpassing a 7B fully fine-tuned model on the BIRD development set.  For a more detailed analysis, please refer to A2 below. Furthermore, the innovative design of READER can inspire new ideas for code analysis in other languages, such as Python and C/C++.

> **Q2: Detailed analysis of READ-SQL generalizability**

**A2:** We present our experimental findings, demonstrating that READ-SQL consistently outperforms SFT CodeS across various domains, with no signs of overfitting, except on the Spider Test Set at the 3B scale.
To evaluate the generalization capabilities of READ-SQL, we reproduced the results of SFT CodeS on three Spider variants (Spider-Syn, Spider-Realistic, and Spider-DK), allowing for a detailed comparison across different difficulty levels. These results are included in Appendix A.5.9 of the revised paper. For fairness, we report only the Generators and their corresponding EX scores for each dataset.



| Dataset          | Model        | Easy       | Medium     | Hard       | Extra       | All        |
|:-----------------|:-------------|:-----------|:-----------|:-----------|:------------|:-----------|
| Spider-Syn       | SFT CodeS-1B | 80.2       | 68.0       | 58.2       | 40.2        | 64.7       |
| Spider-Syn       | Generator-1B | 77.4(-2.8) | 69.5(+1.5) | 61.0(+2.8) | 39.6(-0.6)  | 65.1(+0.4) |
| Spider-Syn       | SFT CodeS-3B | 83.4       | 74.8       | 67.8       | 58.0        | 73.1       |
| Spider-Syn       | Generator-3B | 84.3(+0.9) | 76.1(+1.3) | 70.6(+2.8) | 56.2(-1.8)  | 73.9(+0.8) |
| Spider-Realistic | SFT CodeS-1B | 88.1       | 77.3       | 63.6       | 41.2        | 70.1       |
| Spider-Realistic | Generator-1B | 87.2(-0.9) | 79.8(+2.5) | 69.7(+6.1) | 42.3(+1.1)  | 72.2(+2.1) |
| Spider-Realistic | SFT CodeS-3B | 95.4       | 84.2       | 70.7       | 57.7        | 78.9       |
| Spider-Realistic | Generator-3B | 93.6(-1.8) | 86.2(+2.0) | 74.7(+4.0) | 62.9(+5.2)  | 81.1(+2.2) |
| Spider-DK        | SFT CodeS-1B | 79.1       | 69.5       | 51.4       | 40.0        | 63.2       |
| Spider-DK        | Generator-1B | 79.1(+0.0) | 70.7(+1.2) | 51.4(+0.0) | 50.5(+10.5) | 65.8(+2.6) |
| Spider-DK        | SFT CodeS-3B | 82.7       | 76.8       | 55.4       | 54.3        | 70.7       |
| Spider-DK        | Generator-3B | 82.7(+0.0) | 76.4(-0.4) | 62.2(+6.8) | 46.7(-7.6)  | 69.9(-0.8) |

Our findings are as follows:


- The Generator generally outperforms SFT CodeS on medium and hard difficulty levels.
- On easy and extra-hard levels, its performance is occasionally weaker than SFT CodeS.
- A significant performance gap between READ-SQL and SFT CodeS is observed in Spider-DK.
- These results indicate that adding extra training data significantly benefits the model on medium-difficulty tasks but may slightly hinder performance on simpler or extremely complex tasks.  More analysis can be explored in the future work.
- This further suggests that READ-SQL leverages additional training data more effectively on challenging datasets, such as BIRD, compared to simpler ones like Spider.

---

### Author Response · Authors · 2024-11-25
**Global Comment (part 2)**

> **Q2: Detailed analysis of READ-SQL generalizability (part 2)**

In the reproduction process, we adopt open-source check-points and the code but observe that the performance of SFT CodeS does not fully align with the results reported in the paper. We share the inference logs in the supplementary material. Below is the updated results (only EX):

| model        | BIRD Dev | Spider Dev | Spider Test | Spider-Syn   | Spider-Realistic | Spider-DK    |
| :----------- | :----------- | :------------- | :-------------- | :----------- | :--------------- | :----------- |
| SFT CodeS-1B | 49.54        | 77.80          | 77.50           | 64.70        | 70.10            | 62.80        |
| Generator-1B | 51.76(+2.22) | 80.20(+2.40)   | 77.00(-0.50)    | 65.10(+0.40) | 72.20(+2.10)     | 65.80(+3.00) |
| READ-SQL-1B  | 52.87(+3.33) | 80.70(+2.90)   | 78.80(+1.30)    | 65.00(+0.30) | 71.50(+1.40)     | 65.60(+2.80) |
| SFT CodeS-3B | 55.02        | 83.30          | 81.90           | 73.10        | 78.90            | 70.30        |
| Generator-3B | 56.98(+1.96) | 83.80(+0.50)   | 80.90(-1.00)    | 73.90(+0.80) | 81.10(+2.20)     | 69.90(-0.40) |
| READ-SQL-3B  | 57.37(+2.35) | 84.20(+0.90)   | 81.20(-0.70)    | 74.00(+0.90) | 80.30(+1.40)     | 71.60(+1.30) |
| SFT CodeS-7B | 57.00        | 85.40          | 83.50           | 74.80        | 82.30            | 72.90        |

Our findings suggest the following:

- A standalone Generator typically outperforms the SFT CodeS.
- At the 1B scale, on three Spider variants, incorporating a Corrector does not surpass the performance of the Generator alone but still achieves higher results than the SFT CodeS.
- In out-of-domain scenarios, READ-SQL exhibits minimal or slightly negative performance gains, while its improvements are more pronounced on in-domain datasets.
- READ-SQL does not outperform SFT CodeS on the Spider Test.
- On the BIRD Dev, READ-SQL surpasses SFT CodeS-7B. Notably, even with just the Generator, READ-SQL-3B demonstrates competitive performance, trailing the SFT CodeS-7B by only 0.02%.

The following problems exist in the above phenomenon:
- Why is Corrector less effective than Generator on the Spider-related datasets ?

We compare the average input length of Corrector in the BIRD Dev, Spider Dev, Spider Test and related variants. The results are as follows:


|         | BIRD Dev | Spider Dev  | Spider Test | Spider-Syn   | Spider-Realistic | Spider-DK    |
| :----------- | :----------- | :------------- | :-------------- | :----------- | :--------------- | :----------- |
| Input length | 1468        | 697          | 712           | 738        | 663            | 792

The results indicate that, due to the lack of external information in the Spider-related datasets, such as column descriptions and column value details, Corrector has limited information to utilize, leading to occasional misjudgments. Corrector performs less effectively on the Spider-related datasets compared to the BIRD dataset.


> **Q3: Re-explaining the reasoning process of READ-SQL**

**A3:** The main text lacks a detailed explanation of READ-SQL's reasoning process. We have added a detailed reasoning process in lines 304-323 of the revised version. Here, we provide supplementary information:

READ-SQL's reasoning unfolds in two key stages: SQL generation and self-correction.
SQL generation :
- Step 1: We adopt the method proposed by CodeS[1] for schema linking and to structure the input for the Generator model.
- Step 2: A special prefix token, [SQL], is appended to the input, prompting the Generator model to produce SQLs.

Self-Correction process is shown in Figure 4:
- Step 1: Parsable SQL in the results are parsed one-by-one by READER into a clause set.
- Step 2: By comparing the clause set of each SQL, we identify high-confidence and low-confidence clauses.
- Step 3: The high-confidence clauses are combined to form the basic SQL.
- Step 4: Table names, column names, and values are extracted from low confidence clauses by Elasticsearch to retrieve relevant database information, including table schema, foreign keys, and value matching details.
- Step 5: The final input is constructed following the method shown in Figure 4 and fed to the Corrector.
- Step 6: The first executable SQL generated by the Corrector is selected as the final result.



reference:

[1] Haoyang Li, Jing Zhang, Hanbing Liu, Ju Fan, Xiaokang Zhang, Jun Zhu, Renjie Wei, Hongyan Pan, Cuiping Li, and Hong Chen. Codes: Towards building open-source language models for text-to-sql. Proc. ACM Manag. Data, 2(3):127, 2024b. doi: 10.1145/3654930. URL https://doi.org/10.1145/3654930

---

### Comment · Area_Chair_aAf5 · 2024-11-25
**Action Required: Respond to Author Rebuttals - Nov 27**

Dear ICLR Reviewers,

The author discussion phase is ending soon. Please promptly review and respond to author rebuttals for your assigned papers. Your engagement is critical for the decision-making process.

Deadlines:
- November 26: Last day for reviewers to ask questions to authors.
- November 27: Last day for authors to respond to reviewers.
- November 28 - December 10: Reviewer and area chair discussion phase.

Thank you for your timely attention to this matter.

---

### Meta-Review · Area_Chair_aAf5 · 2024-12-21

**Metareview:**

The paper proposes READ-SQL, a framework that improves text-to-SQL performance through SQL decomposition and self-correction mechanisms. The framework utilizes AST parsing to decompose SQL queries into components and reasoning paths, employing a Generator for initial SQL formulation and a Corrector for refinement based on confidence levels.

Some reviewers acknowledge the framework's strengths in achieving competitive performance with smaller models and its novel approach to data augmentation through SQL decomposition. The results show improvements over baselines on benchmarks like BIRD and Spider, particularly for models under 3B parameters. However, reviewers raise several significant concerns. First, the technical novelty appears incremental, with unclear advantages over existing approaches like schema linking and end-to-end methods. Second, the experimental evaluation lacks comparisons with recent advanced baselines, and the performance gains may be attributed to overfitting, particularly on the BIRD dataset. Third, the paper's presentation has notable gaps, especially in explaining the self-correction process and inference pipeline. While the authors provided responses to these concerns, multiple reviewers indicated that their core reservations about technical novelty and contribution remain unaddressed.

Overall, given these fundamental limitations, the work would benefit from substantial improvements before meeting the publication threshold.

**Additional Comments On Reviewer Discussion:**

Check the meta review.

---

### Decision · Program_Chairs · 2025-01-22

Reject